# The Rho-GEF PIX-1 directs assembly or stability of lateral attachment structures between muscle cells

Jasmine C. Moody 🔘 [1], Hiroshi Qadota[1], April R. Reedy[1], C. Denise Okafor[2], Niveda Shanmugan[1], Yohei Matsunaga[1], Courtney J. Christian[1], Eric A. Ortlund 🔘 [2] & Guy M. Benian 🔘 [1✉]

PIX proteins are guanine nucleotide exchange factors (GEFs) that activate Rac and Cdc42, and are known to have numerous functions in various cell types. Here, we show that a PIX protein has an important function in muscle. From a genetic screen in *C. elegans*, we found that *pix-1* is required for the assembly of integrin adhesion complexes (IACs) at borders between muscle cells, and is required for locomotion of the animal. A *pix-1* null mutant has a reduced level of activated Rac in muscle. PIX-1 localizes to IACs at muscle cell boundaries, M-lines and dense bodies. Mutations in genes encoding proteins at known steps of the PIX signaling pathway show defects at muscle cell boundaries. A missense mutation in a highly conserved residue in the RacGEF domain results in normal levels of PIX-1 protein, but a reduced level of activated Rac in muscle, and abnormal IACs at muscle cell boundaries.

[1] Department of Pathology, Emory University, Atlanta, GA 30322, USA. [2] Department of Biochemistry, Emory University, Atlanta, GA 30322, USA.
✉email: pathgb@emory.edu

In striated muscle, sarcomeres are attached end to end to create myofibrils that extend the length of the elongated muscle cell. The myofibrils are tightly packed and connected to each other via intermediate filaments. Myofibrils in both skeletal and cardiac muscle cells are connected at the periphery of the muscle cell to the cell membrane and extracellular matrix (ECM) via costameres, muscle-specific integrin adhesion complexes (IACs). Although detectable beneath the entire sarcolemma of skeletal muscle, the dystrophin-glycoprotein complex (DGC) is enriched at costameres[1]. Genetic deficiencies of dystrophin and a number of DGC proteins result in muscular dystrophies. Deficiency of costameric component integrin α7, results in a congenital myopathy[2]. Heterozygous mutations in several other costameric proteins result in dilated and hypertrophic cardiomyopathy including vinculin, α-actinin-2, four and a half LIM protein-2 (FHL2), and ILK[3].

*C. elegans* is an excellent genetic model organism in which to learn new principles of sarcomere assembly, maintenance, and regulation[4]. The major striated muscle of *C. elegans* is found in the body wall and is required for locomotion. Similar to striated muscle in other animals, the thin filaments are attached to Z-disks (called dense bodies), and the thick filaments are attached to M-lines. However, the myofibrils are restricted to a narrow ~1.5 μm zone adjacent to the cell membrane along the outer side of the muscle cell, and all the dense bodies and M-lines are anchored to the muscle cell membrane and ECM. This architecture, together with what is known about the molecular composition of these structures, indicates that nematode dense bodies and M-lines also act as costameres. *C. elegans* body wall muscle also has IACs at the muscle cell boundaries[5], where they form attachment plaques that anchor the muscle cell to a thin layer of ECM that lies between adjacent muscle cells.

IACs (aka focal adhesions or adhesomes) consist of the transmembrane protein integrin and hundreds of different proteins in a complex both in the ECM and especially intracellularly[6–9]. IACs are important for many cell types. The adhesion of cells to a matrix is crucial for both tissue formation and for cell migration. In stationary cells like muscle, these complexes are rather stable, but in motile cells they are dynamic, with new complexes assembled at the leading edge and older complexes disassembled at the trailing edge[6]. Studies of platelets, leukocytes, and tissue culture cells indicate that normally when integrins are expressed on the cell surface they are in a compact or bent or inactive state, unable to bind to their extracellular targets, but can become "activated" to bind via several triggers (chemokine to chemokine receptor interaction, local increase in PIP2 or calpain) that lead to binding of the cytoplasmic tail of β-integrin to talin. Talin binding drives integrin to a more open conformation, competent to bind extracellular targets[10]. Kindlin, which is similar in domain composition to talin, is also involved in integrin activation by clustering of talin-activated integrins, at least in mammalian platelets[11]. Although we understand the steps involved in the formation of IACs[7–9], we do not know how the composition of an IAC is determined, and we do not know what determines where an IAC forms. This question is especially important for skeletal muscle cells, in which one type of IAC (the costamere) forms at regular intervals and anchors the peripherally located myofibrils to the sarcolemma and ECM at the level of Z-disks.

Here, we show that through a genetic screen in *C. elegans*, identification of a signaling molecule, PIX-1 (orthologous to β-PIX in mammals), that is required for assembly or stability of IACs only at muscle cell boundaries. PIX proteins consist of SH3 and RhoGEF domains, and a coiled-coil region, and are known to act as guanine exchange factors (GEFs) for activation of the small GTPases, Rac1 and/or Cdc42. Antibodies to PIX-1 localize to all three types of muscle IAC-muscle cell boundaries, M-lines and dense bodies. A biochemical pathway for PIX-1 can be inferred from studies of PIX proteins in mammals and *pix-1* in several non-muscle tissues in *C. elegans*. Analysis of loss of function mutants for genes encoding proteins at all known steps of this pathway show defects of muscle cell boundaries similar to that of *pix-1* mutants. As compared to wild type, a *pix-1* null mutant and a *pix-1* missense mutant show reduced levels of activated (GTP bound) Rac in muscle. Both deficiency and overexpression of wild-type PIX-1 protein result in disrupted muscle cell boundaries and decreased whole nematode locomotion, suggesting that the level of PIX-1 protein needs to be tightly regulated. Our results demonstrate that the PIX signaling pathway has an important function in muscle.

## Results

**pix-1 mutants lack IACs at muscle cell boundaries.** We screened the Million Mutation Project (MMP)[12] mutant strains for defects in integrin adhesion complex organization. After screening 574 strains, we identified one strain, VC20386, that by immunostaining showed lack of localization of one IAC component, PAT-6 (α-parvin), at muscle cell boundaries, but normal localization of PAT-6 at M-lines and dense bodies (Fig. 1a). Using a combination of outcrossing to wild-type and SNP mapping, we determined that this phenotype is due to a nonsense mutation in a single gene, *pix-1* (see "Methods"), with allele designation gk299374. We then obtained six additional strains from the *Caenorhabditis* Genetics Center (CGC) that contained mutations in *pix-1*. Two are intragenic deletions, one is a nonsense mutation, and three are missense mutations. Except for the deletions, these additional *pix-1* alleles come from the MMP. WormBase indicates that there are 17 MMP alleles of *pix-1*, but the ones we chose to study either have nonsense mutations (the original one from our immuno-screening, and one more), or have missense mutations residing in conserved protein domains (see below for PIX-1 domain structure). By immunostaining, we found the absence of PAT-6 at the muscle cell boundaries in four out of six of these additional strains (Fig. 1b; Supplementary Fig. 1). The two intragenic deletion alleles, gk416 and ok982, and the additional nonsense allele, gk299384, show the same boundary defect phenotype as the original allele, gk299374, thus confirming that loss of function of *pix-1* is responsible for the phenotype. One of the three missense alleles, gk893650, shows a similar phenotype, but two of the missense alleles, gk406361 and gk713465, display a wild-type boundary phenotype. Perhaps these two missense mutants, gk406361 and gk713465, showing no phenotype are affecting residues in PIX-1 that are not essential for function. Indeed, gk713465 is an R212Q substitution in a residue that is not conserved in the RhoGEF domain (see below).

Additional evidence that *pix-1* is responsible for the muscle boundary phenotype, is that we were able to rescue the phenotype by expressing a wild-type *pix-1* cDNA under the control of a muscle-specific promoter (Fig. 1c). By immunostaining, we found that in the nonsense mutant, *pix-1(gk299374)*, other IAC components are missing from the muscle cell boundaries, including UNC-52 (perlecan) in the ECM, PAT-3 (β-integrin) at the muscle cell membrane, UNC-112 (kindlin), and UNC-95 (Fig. 1d). Thus, we conclude that PIX-1 is required for the assembly or stability of IACs at the muscle cell boundary.

The standard view of integrin adhesion sites is that they act as a physical link between the ECM and cell membrane to cortical actin filaments. Like other cells, muscle cells are known to have cortical actin, however, this has not been previously reported for *C. elegans* muscle cells. Our attempts to visualize actin filaments near the muscle cell boundary using phalloidin staining were not

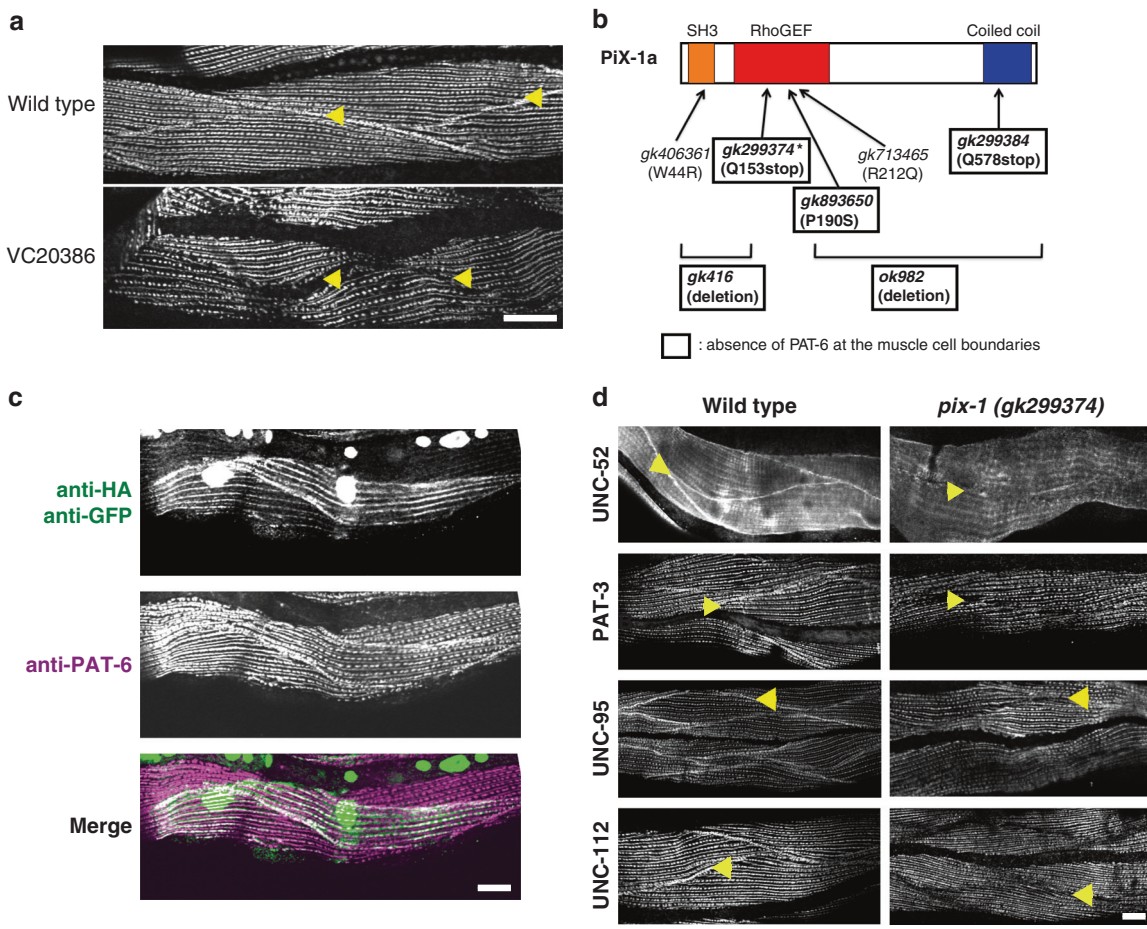

**Fig. 1 Identification of *pix-1* as a gene required for the assembly of IACs at muscle cell boundaries. a** Confocal images of several body wall muscle cells immunostained with antibodies to PAT-6 (α-parvin) from wild type and the strain VC20366 identified by screening 574 MMP strains. Arrowheads point to the boundaries between muscle cells. **b** Schematic representation of domains in *C. elegans* PIX-1a, and the location and nature of 7 *pix-1* mutants and evaluation of their phenotypes. The asterisk denotes the *pix-1* mutant allele found in the original strain VC20366. **c** Muscle-specific expression of a wild-type cDNA for PIX-1 tagged with HA rescues the phenotype of *pix-1(gk299374)*. The transgene is *sfEx61*[*myo-3p::HA-PIX-1; sur-5::nls::GFP*], in which sur-5::nls::GFP is the transformation marker showing GFP in nuclei. Note that PAT-6 has been restored to the muscle cell boundaries (indicated by arrowheads), and that HA-tagged PIX-1 localizes to muscle cell boundaries, dense bodies, and M-lines. **d** Comparison of wild type vs. *pix-1(gk299374)* immunostained with antibodies to the indicated IAC proteins and imaged by confocal microscopy. Arrowheads denote muscle cell boundaries. Note that all four proteins are present in wild type but missing from muscle cell boundaries in the *pix-1* mutant. Each image is a representative image obtained from at least 2 fixation and immunostaining experiments, and imaging at least three different animals. Scale bars in (**a**), (**c**) and (**d**), 10 μm.

successful, for unknown reasons. As an alternative, we imaged F-actin using strain KAG547 that expresses in body wall muscle both GFP-myosin (MHC A) and LifeAct-mCherry. LifeAct is a 17 amino acid peptide that binds to F-actin and does not interfere with actin dynamics[13]. In wild-type animals, as shown in Fig. 2 (top row), there is clearly a thin band of F-actin that lies near the muscle cell boundary (indicated by yellow arrow), which can be identified by three criteria: (1) its location between two adjacent spindle-shaped body wall muscle cells (Supplementary Fig. 2), (2) being thinner than a typical I-band which alternates with myosin A-bands, and (3) not projecting throughout the depth of the myofilament lattice[5] like a typical I-band, which can be discerned by observing less intense signal when the optical slice is taken deeper into the lattice (the label of deeper part in Fig. 2 and Supplementary Fig. 2). If there is cortical actin underneath the cell membranes of each adjacent muscle cell, why do we not observe two closely spaced lines? The likely reason is that the cells are very close to each other and these lines are not resolvable by the light microscope. We crossed the KAG547 strain into *pix-1 (gk299374)*. In this *pix-1* nonsense mutant, the muscle cell boundaries clearly have two separated F-actin lines at the muscle

cell boundary (bottom two rows in Fig. 2 and Supplementary Fig. 2). Therefore, although *pix-1* is required for assembly or maintenance of IACs at the muscle cell boundary, it is not required for the assembly of cortical F-actin to which it likely interacts. The fact that two clearly separated F-actin lines can be observed in the *pix-1* nonsense mutant indicates that the cells are separated from each other, likely due to less adhesion to the ECM lying between adjacent cells.

**Nematode PIX-1 is most similar to human β-PIX.** A BLAST search reveals that *C. elegans* PIX-1 is most similar to mammalian β-PIX and α-PIX. Mammals have 2 PIX proteins, α-PIX and β-PIX, encoded by separate genes, whereas *C. elegans* has a single gene encoding a single PIX protein[14,15]. Our sequence and domain analysis of PIX-1 and human α-PIX and β-PIX proteins indicates that PIX-1 is quite similar to both α-PIX and β-PIX, but more similar to β-PIX because PIX-1, like β-PIX, is missing the CH domain found in α-PIX, and the SH3, RhoGEF and coiled-coil regions of PIX-1 are slightly more identical to those in β-PIX (Fig. 3a).

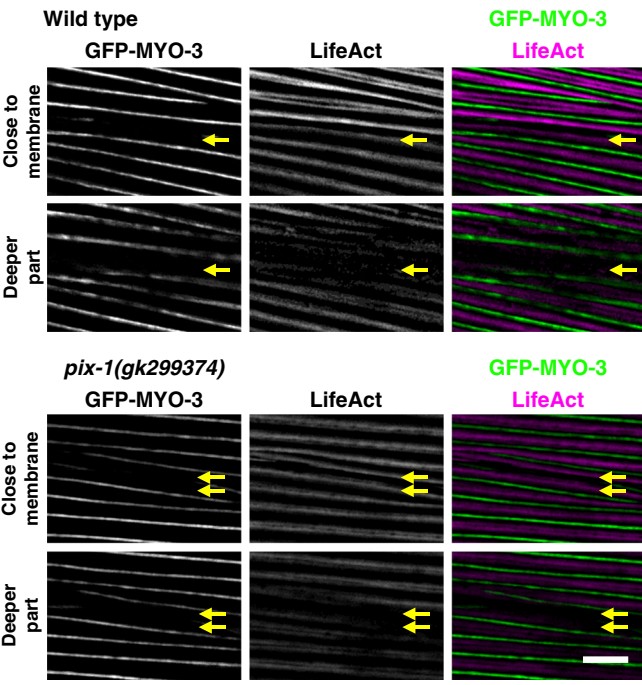

**Fig. 2 Live imaging of cortical F-actin at muscle cell boundaries.** SIM images of portions of two adjacent body wall muscle cells from a nematode strain in which a muscle myosin MHC A was tagged with GFP by CRISPR and LifeAct-mCherry was expressed in muscle cells from a transgene. GFP-MYO-3 (MHC A) labels the middle of sarcomeric A-bands, and LifeAct-mCherry labels I-bands, except for F-actin at the boundary between two adjacent muscle cells (indicated by yellow arrows). Note how the signal from the F-actin at the boundary diminishes as the focal plane changes from close to the outer muscle cell membrane to deeper into the myofilament lattice whereas the F-actin signal from I-bands does not change. Also note that in the *pix-1* nonsense mutant, *gk299374*, there are two bands of cortical F-actin at the boundary. Each image is a representative image obtained from imaging at three different animals of each strain. Scale bar, 5 μm.

**PIX-1 acts as a GEF for the Rac, CED-10, in muscle**. PIX proteins are known to act as guanine nucleotide exchange factors (GEFs) for Rac1 and/or Cdc42. *C. elegans* has three Rac proteins encoded by separate genes, CED-10, MIG-2, and RAC-2. As shown later, analysis of mutants in these proteins indicate that only CED-10 is required for the assembly of IACs at muscle cell boundaries. Therefore, we asked whether the activation of CED-10 might be defective in a *pix-1* loss of function mutant. To address this question, we generated a transgenic line in which HA-tagged CED-10 was expressed in body wall muscle using the muscle-specific *myo-3* promoter. We adapted the use of a commercially available kit for Rac1 activation to measure the status of CED-10 activation, that is based on the ability of GST-PAK-PBD to pull down activated or GTP-bound CED-10, but not inactive or GDP-bound CED-10. To validate our assay, worm lysates were incubated with an excess of GDP or the non-hydrolysable GTPγS, and the amount of activated CED-10 was compared by western blot using anti-HA. As shown in Fig. 3b, much more activated CED-10 was pulled down with GTPγS than with GDP. Our HA-CED-10 muscle-expressed reporter strain was then crossed into *pix-1(gk299374)*. As shown in Fig. 3c, less activated CED-10 was pulled out from *pix-1(gk299374)* than from wild type. After repeating this experiment three times, the mean level of activated CED-10 from *pix-1(gk299374)* was 54.0 +/− 7.6% (mean and standard deviation) of the level from wild type. As shown in Fig. 6b, *pix-1(gk299374)*, shows no detectable PIX-1 protein by

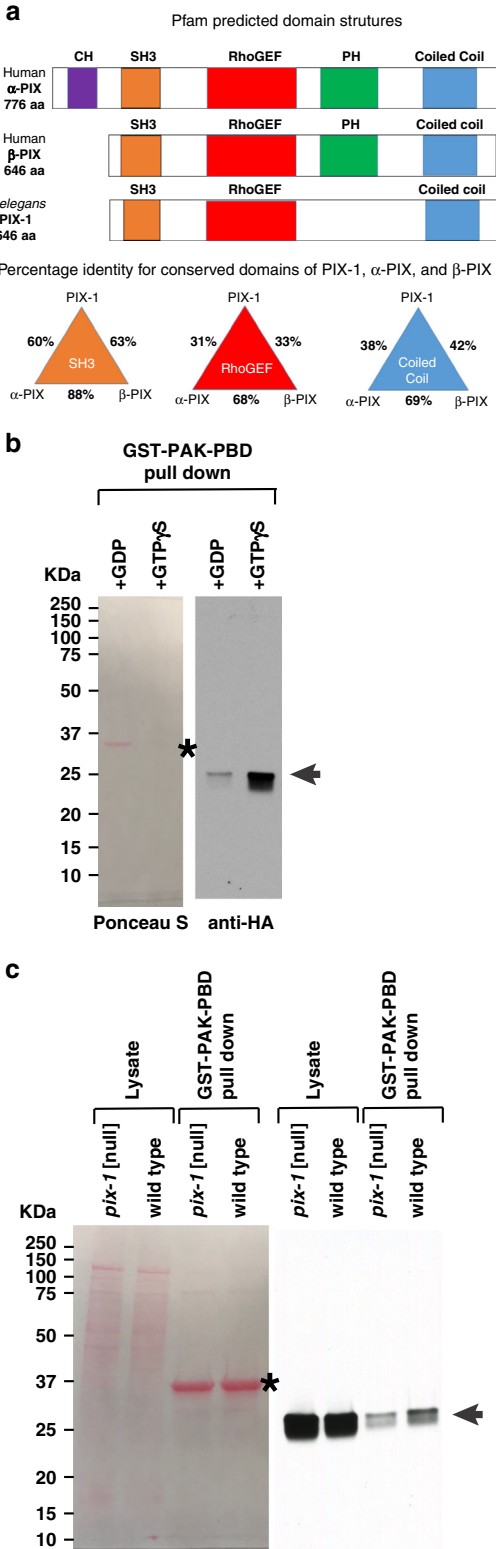

western blot and is thus likely to be a null mutant. That the amount of activated CED-10 is not zero in this mutant likely reflects the expression of other Rac GEFs in *C. elegans* muscle, one of these known to be UNC-73 (TRIO)[16,17]. Nevertheless, these results indicate that PIX-1 is a GEF for the Rac, CED-10, in body wall muscle.

**Fig. 3 PIX-1 is most similar to human β-PIX and acts as a GEF for CED-10 (Rac) in muscle. a** Schematic representation of predicted domains in *C. elegans* PIX-1a, human α-PIX and human β-PIX. The bottom triangles show a comparison of percentage identities for SH3, RhoGEF, and coiled-coil regions. **b** Validation of activation assay. Lysates were prepared from nematodes expressing HA-CED-10 in body wall muscle using a muscle-specific promoter, an excess of GDP or GTPγS, was added, and then beads coupled to GST-PAK-PBD were used to pull down activated CED-10 (i.e. bound to GTP or GTP + GTPγS). Samples were separated on a gel, blotted and incubated with antibodies to HA. The asterisk indicates the position of GST-PAK-PBD on the blot. Arrow indicates the position of HA-CED-10●GTP/GTPγS on the western blot. **c** Deficiency of PIX-1 results in a reduced level of activated CED-10. Lysates were prepared from two strains, each expressing HA-CED-10 in body wall muscle: wild type, and *pix-1 (gk299374)*, a nonsense mutant that results in no detectable PIX-1. These lysates were incubated with beads coupled to GST-PAK-PBD, and used to pull down activated CED-10. Both the Ponceau S stained blot, and the result of the western using anti-HA are shown. The western shows, from left to right: total HA-CED-10 in *pix-1* mutant, total HA-CED-10 in wild type, HA-CED-10●GTP in *pix-1* mutant, and HA-CED-10●GTP in wild type. Asterisk indicates the position of GST-PAK-PBD on the blot. Arrow indicates the position of HA-CED-10 from the lysate, or HA-CED-10●GTP from the pulldown.

**Similar phenotype from deficiency or overexpression of *pix-1*.**
In *C. elegans* muscle, the force of muscle contraction that bends the worm and thus propels locomotion (swimming, crawling, burrowing), is transmitted through all three integrin attachment sites, the M-lines, the dense bodies and the adhesion plaques at the muscle cell boundaries. Thus, mutants that are defective in these structures, in many cases, show reduced whole-animal locomotion[18]. As shown in Fig. 4a and b, the *pix-1* nonsense mutant, *gk299374*, the *pix-1* intragenic deletion, *gk416* (each outcrossed 5× to wild type) and the *pix-1* intragenic deletion, *ok982* (outcrossed 3X to wild type), show reduced locomotion in swimming in buffered water and in crawling along an agar surface. However, *pix-1(gk893650)* (outcrossed 5× to wild type) which has the missense mutation P190S, and has a more subtle boundary defect (see below), displays normal swimming and crawling motility (Fig. 4a, b). We next wondered whether the *pix-1* rescued strain would show normal or near normal locomotion. To our surprise, in both swimming and crawling, the integrated transgene expressing wild-type *pix-1* cDNA from the muscle-specific promoter for *myo-3*, [*pix-1(gk299374); sfIs20*], was slower than wild type (Fig. 4c, d). One possibility is that the integration occurred in a gene essential for normal locomotion, for example, a muscle or neuronal Unc gene. However, this does not appear to be the explanation for slow movement: Slow movement was observed even in a strain in which the extrachromosomal array was expressed in a wild-type background, [wild type; *sfEx61*] (Fig. 4c, d). Thus, the most likely explanation for reduced motility shown by the strains carrying the transgene, is overexpression of PIX-1, which is typical for extrachromosomal or integrated arrays. Indeed, quantitative western blotting using an antibody to PIX-1 (described below) shows that the integrated array expresses six times the amount of PIX-1 as found in wild type (Fig. 5a, b). In addition to a motility defect, overexpression of PIX-1 results in a defective muscle cell boundary (Fig. 5c). Finally, although we found a motility defect in three independently generated loss of function mutants, we were concerned that the motility defect might result from mutation in a gene closely linked to *pix-1*. In order to eliminate this possibility, we used CRISPR/Cas9 to correct the TAA stop codon in *pix-1(gk299374)*, to the wild-type sequence of a CAA Gln codon. As shown in Fig. 4c, d, this strain,

*pix-1(syb2137gk299374)* displays normal swimming and crawling motility, and as shown in Fig. 5d, normal muscle cell boundaries. We conclude that either loss of function or overexpression of *pix-1* results in reduced locomotion and a muscle cell boundary defect.

**_pix-1_ mutants have normal organization of muscle sarcomeres.**
The sarcomeres of *pix-1* mutants are normally organized: thin filaments (phalloidin), thick filaments (anti-MHC A), dense bodies (anti-α-actinin), and M-lines (anti-UNC-89 (obscurin)) show the same localization in *pix-1(gk299374)* as they do in wild-type muscle (Supplementary Fig. 3). Therefore, the defects in locomotion in *pix-1* mutants might be attributed to a defect in force transmission through a lack of IACs at the lateral muscle cell boundaries, thus demonstrating both the functional importance of these structures, and the importance of PIX-1 in establishing or maintaining these structures.

**PIX-1 localizes to M-line, dense body, and adhesion plaque.**
We developed two sets of polyclonal antibodies to PIX-1. WormBase predicts that *pix-1* encodes two protein isoforms, PIX-1a (646 residues) and PIX-1b (450 residues) (Fig. 6a). The first immunogen chosen (#1) was expected to generate antibodies that could detect both of these isoforms. The resulting rabbit polyclonal antibodies were of low titer and allowed western blot detection of PIX-1a but not PIX-1b. Also, these antibodies to immunogen #1 failed to localize in muscle by immunostaining experiments. We then generated antibodies to a second immunogen (#2). As shown in Fig. 6b, this higher-titer antibody detected a protein of ~80 kDa, close to the expected size for PIX-1a (73.2 kDa) from wild type, but not from the nonsense or two intragenic deletion alleles of *pix-1*. Interestingly, these antibodies detect a PIX-1 protein of normal size and abundance from *pix-1 (gk893650)*, which expresses PIX-1 with the missense mutation P190S in the RhoGEF domain. The expression of normal levels of intact PIX-1 protein likely explains why the muscle cell boundary defect is more subtle in *pix-1(gk893650)*. (Supplementary Figs. 1; Fig. 6c; Fig. 9b, c).

We next used these anti-PIX-1 antibodies (to immunogen #2) to immunostain body wall muscle. As shown in Fig. 6c, anti-PIX-1 localizes to muscle cell boundaries, to M-lines and dense bodies, and this staining is not detectable in the *pix-1* nonsense, *gk299374*, and intragenic deletion, *gk416*, mutants. However, consistent with the immunoblot results anti-PIX-1 staining is detectable at muscle cell boundaries in *pix-1(gk893650)* P190S.

**Determination of the PIX-1 signaling pathway in muscle.**
Based on studies of β-PIX in mammals and some studies of *pix-1* in *C. elegans* (but not in muscle), we hypothesized that PIX-1 functions in the biochemical pathway shown in Fig. 7a: that it activates a Rac/Cdc42 family member, perhaps via the scaffold protein GIT-1 (or additional or another scaffold protein), and this Rac/Cdc42 family member acts through a PAK protein kinase. *C. elegans* has 3 Rac proteins, CED-10, MIG-2, and RAC-2; one Cdc42, CDC-42; and 3 PAK protein kinases, PAK-1, PAK-2, and MAX-2. There are loss or reduction of function mutants available for all of the proteins indicated in Fig. 7a, except for CDC-42. As shown in Fig. 7b (and summarized in Fig. 7a), loss of function mutants in GIT-1, CED-10, PAK-1, and PAK-2, but not MIG-2, RAC-2 or MAX-2, result in the absence or reduced accumulation of PAT-6 at muscle cell boundaries. The defect in *pak-2(ok332)* is more subtle than the other mutants (Fig. 7b, bottom row); although there is not complete absence of PAT-6, PAT-6 appears less concentrated or more discontinuous at the cell boundaries than it does in wild type. Also, as shown in Supplementary Fig. 4,

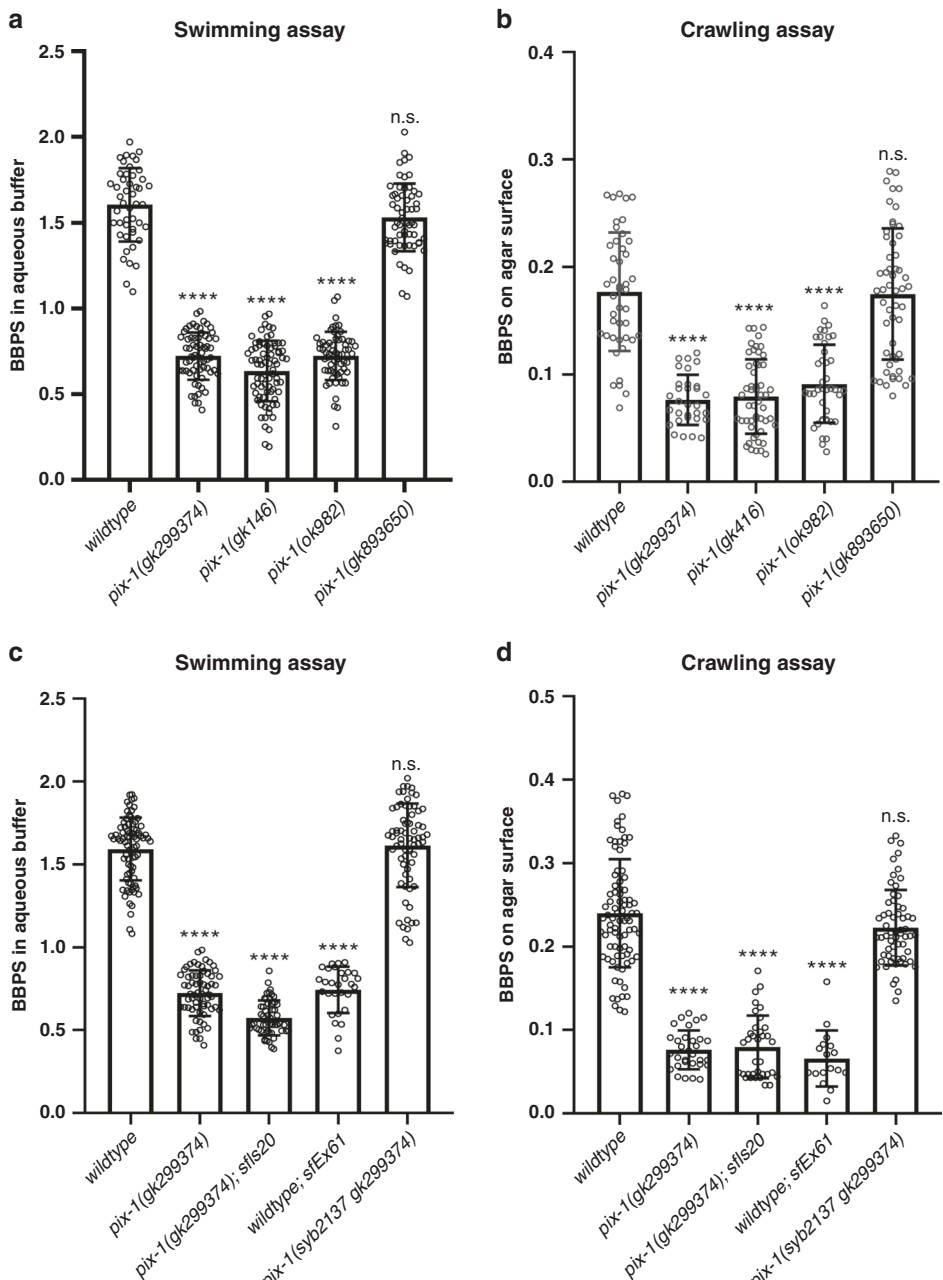

**Fig. 4 Both loss of function and overexpression of *pix-1* results in reduced locomotion. a** Swimming and **b** crawling assays show that loss of function mutations in *pix-1* result in reduced locomotion compared to wild type. **c** Swimming and **d** crawling assays show that both the integrated rescued strain, *pix-1(gk299374); sfIs20*, and wild type expressing the rescuing transgene from an extrachromosomal array, *sfEx61*, have reduced locomotion. In contrast, CRISPR/Cas9 repair of the nonsense mutation in *pix-1(gk299374)*, called *pix-1(syb2137 gk299374)*, results in normalization of locomotion. In the graphs, each open circle represents the result from an independently selected animal. The exact n values vary, and these data can be found in the Source Data Files. Student's two-sided *t* test was used to test for significance. Error bars: standard deviations; ****$p \leq 0.0001$; n.s.: no significant difference.

we obtained similar results for an additional allele of *rac-2*, *gk281*, and an additional allele of *ced-10*, *n1993*. However, the disruption of PAT-6 organization at muscle cell boundaries is less severe for *ced-10(n1993)* than it is for *ced-10(n3246)*. This is consistent with the nature of the mutations: *ced-10(n3246)* is a G60R mutation at a highly conserved residue in the middle of the protein, whereas *ced-10(n1993)* is a V128G mutation at the penultimate residue that is only moderately conserved.

We next asked whether the level of PIX-1 protein might be affected by the deficiency of other pathway proteins. As shown in Fig. 8, using anti-PIX-1 in a quantitative western, we found that in either *git-1(ok1848)* or *pak-1 (ok448)*, but not *ced-10(n3246)*,

there are reduced levels of PIX-1 as compared to wild type. These data suggest that GIT-1 and PAK-1 are required for PIX-1 stabilization.

**A boundary defect from mutation P190S in the RhoGEF domain.** PFAM alignment of RhoGEF domains from PIX proteins across 10 different species shows that P190 is absolutely conserved (Fig. 9a). This conservation suggests that P190 is required for RhoGEF activity, and that the P190S missense mutation results in reduced RhoGEF activity. Our alignment of these RhoGEF domains also indicates that PIX-1 R212 is not

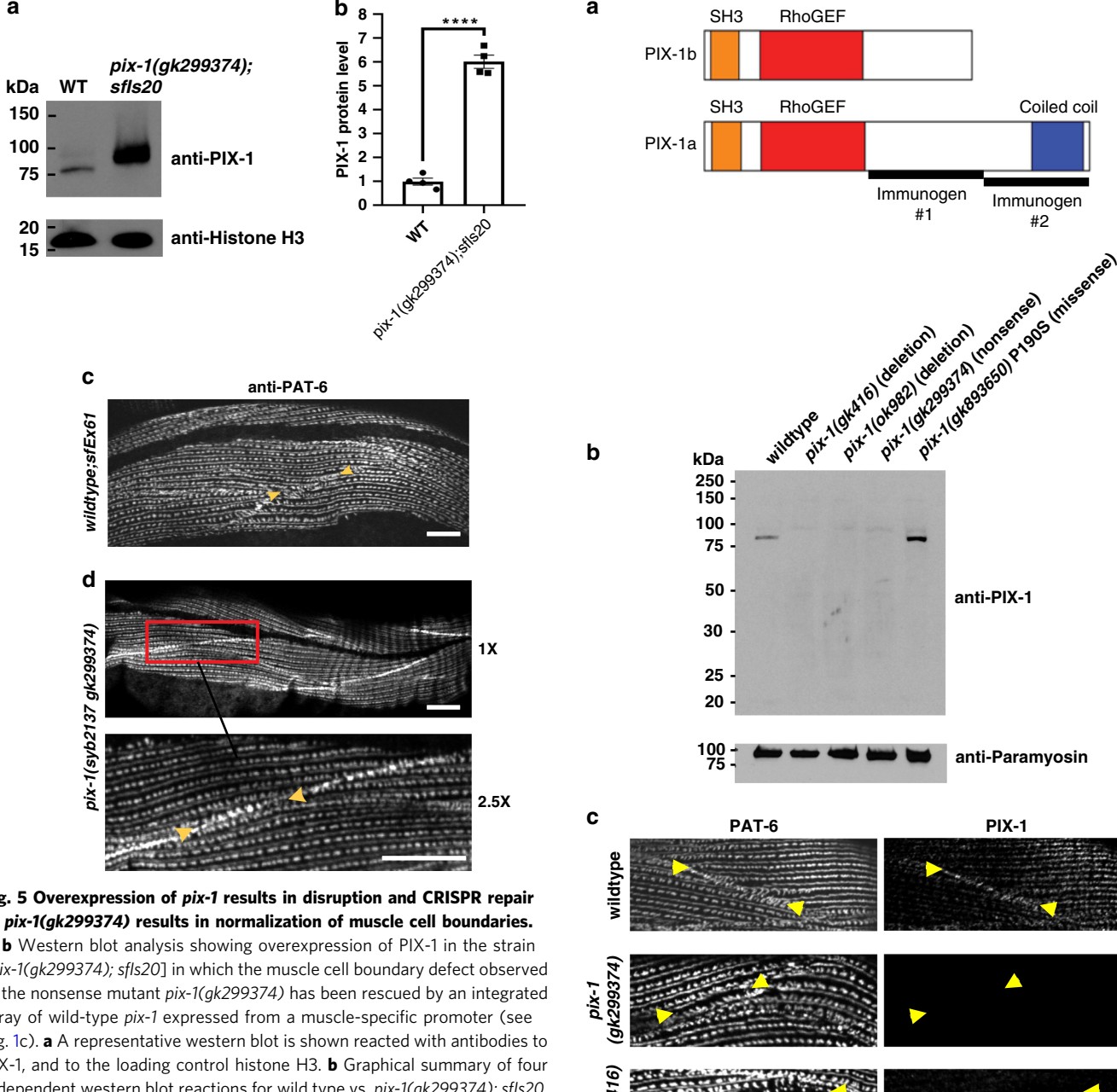

**Fig. 5 Overexpression of *pix-1* results in disruption and CRISPR repair of *pix-1(gk299374)* results in normalization of muscle cell boundaries. a**, **b** Western blot analysis showing overexpression of PIX-1 in the strain [*pix-1(gk299374); sfIs20*] in which the muscle cell boundary defect observed in the nonsense mutant *pix-1(gk299374)* has been rescued by an integrated array of wild-type *pix-1* expressed from a muscle-specific promoter (see Fig. 1c). **a** A representative western blot is shown reacted with antibodies to PIX-1, and to the loading control histone H3. **b** Graphical summary of four independent western blot reactions for wild type vs. *pix-1(gk299374); sfIs20*. Means and standard errors of the means are shown. The two strains show statistically different levels of PIX-1 using a two-sided students *t*-test with *p* < 0.0001 (indicated by ****). Images of the entire western blots are provided in the Source Data Files. **c** This same array when expressed in a wild-type background [wild-type; *sfEx61*] disrupts the muscle cell boundary. **d** *pix-1(syb2137gk299374)* is *pix-1(gk299374)* after CRISPR/Cas9 was used to repair the nonsense mutation. Confocal images of anti-PAT-6 are shown and reveal that the muscle cell boundary is normal. This is further evidence that the muscle cell boundary defect is due specifically to mutation in the *pix-1* gene. Arrowheads point to muscle cell boundaries visualized by immunostaining with anti-PAT-6. Each image is a representative image obtained from at least 2 fixation and immunostaining experiments, and imaging at least three different animals. Scale bars, 10 μm.

conserved, and this might explain why *pix-1(gk713465)* R212Q has no obvious phenotype (Fig. 1b, Supplementary Fig. 1). Closer examination of confocal images of *pix-1(gk893650)* shows that the boundary defect is more subtle in this mutant than in the nonsense and intragenic deletion *pix-1* mutants. In Qadota et al.[5], we

reported that by structured illumination microscopy (SIM), in wild-type muscle, the boundaries appear like a zipper, in which the two sides of the zipper are closely apposed to each other (as if the zipper were closed). Zoomed-in confocal views of muscle (Fig. 9b) using antibodies to two different IAC components, UNC-95 and PAT-6 (α-parvin), show a closed zipper in wild type, whereas in *pix-1(gk893650)* the zipper appears open. This result is also revealed by SIM imaging of PAT-6 staining (Supplementary Fig. 5). A similar "open zipper" appears with anti-UNC-112 (kindlin) staining (Fig. 9c), although anti-UNC-52

**Fig. 6 Antibodies to PIX-1 detect PIX-1a on a western blot and localize to muscle cell boundaries, M-lines, and dense bodies. a** Schematic representation of domains in the predicted isoforms PIX-1a and PIX-1b and regions used as immunogens to generate antibodies. **b** Western blot detection of a protein of expected size for PIX-1a from wild type and from *pix-1(gk893650)* [P190S] but not from deletion or nonsense *pix-1* mutants. Anti-paramyosin was used as a gel loading control. An image of the entire blot reacted with anti-paramyosin is available in the Source Data Files. **c** Antibodies to PIX-1 localize to muscle cell boundaries and with less intensity to M-lines and dense bodies. Each strain was co-stained with antibodies to PAT-6 and PIX-1, and imaged by confocal. Note the lack of PIX-1 immunostaining in the nonsense and deletion *pix-1* mutants, but strong and disorganized staining in *pix-1(gk893650)* [P190S]. Arrowheads point to a muscle cell boundary. Each image is a representative image obtained from at least 2 fixation and immunostaining experiments, and imaging at least three different animals of each strain. Scale bar, 10 μm.

(perlecan) and anti-PAT-3 (β-integrin) staining show only less of these proteins at the boundaries of *gk893650* (Fig. 9c). In summary, the immunoblot and immunolocalization results using anti-PIX-1 show that PIX-1 P190S is a stable protein that localizes to the general vicinity of the muscle cell boundaries, but examination of other IAC components show that although each half of the zipper is formed, these halves are abnormally separated from each other. Taken together with the conservation of P190 in the RhoGEF domain, this suggests that RhoGEF activity is required for proper muscle cell boundary organization.

**P190S may alter RhoGEF structure and binding with Rac•GTP.** We predicted the structure of the RhoGEF domain of PIX-1 by homology modeling using the NMR structure of human β-PIX[19] as a template. We also predicted the structure of a complex between PIX-1 and both GDP- and GTP-bound forms of Rac1 GTPase, using GEF-Rac1 complexes as templates. We analyzed the helical conformation in PIX-1 molecular dynamic (MD)

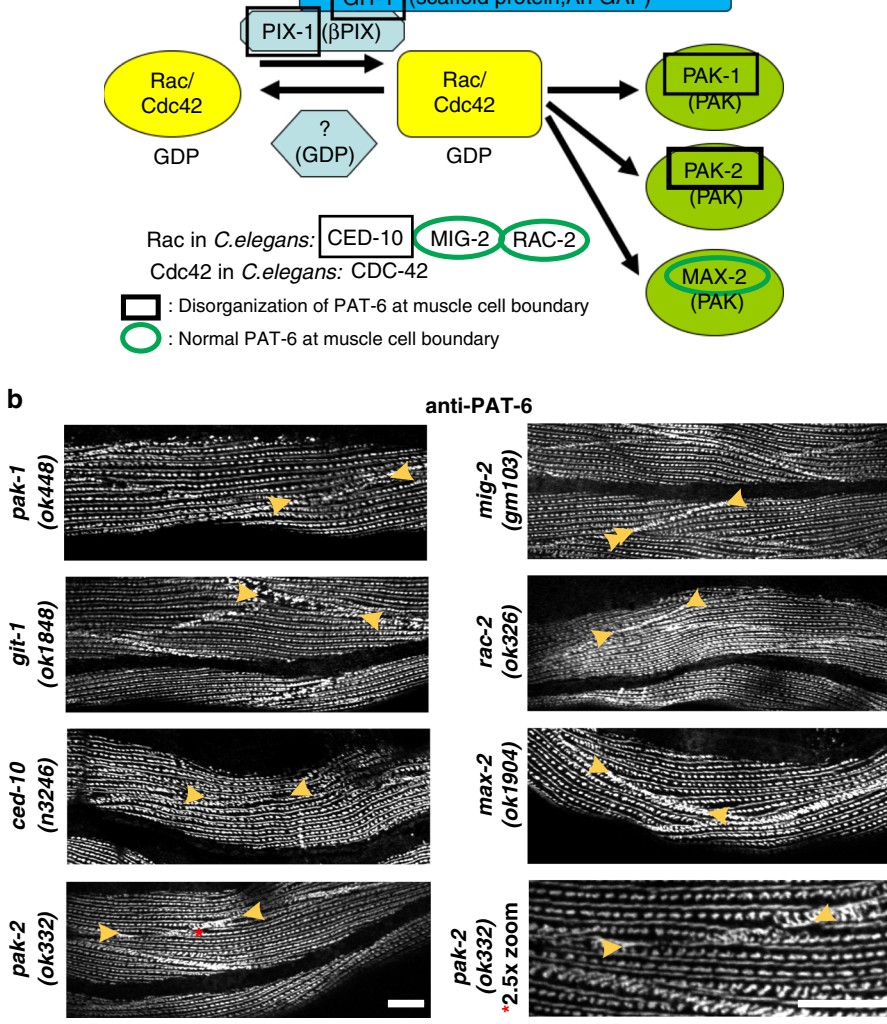

**Fig. 7 Mutations in genes encoding known proteins of a PIX-1 pathway result in muscle cell boundary disruption. a** Putative PIX-1 biochemical pathway based on what is known of PIX proteins in mammals and other cell types in *C. elegans*. Also indicated is a summary of the results shown in **b**. **b** Confocal images of body wall muscle from the indicated mutants immunostained with anti-PAT-6. Note reduced or disorganized PAT-6 localization at muscle cell boundaries (indicated by arrowheads) in *pak-1, git-1, ced-10,* and *pak-2* mutants, but normal PAT-6 localization in *mig-2, rac-2,* and *max-2* mutants. A higher magnification view (bottom row, right) of *pak-2(ok332)* is shown because it has a more subtle defect than the other mutants. Results on second alleles for *rac-2* and *ced-10* are shown in Supplementary Fig 3. Each image is a representative image obtained from at least 2 fixation and immunostaining experiments, and imaging at least three different animals of each strain. Scale bar, 10 μm.

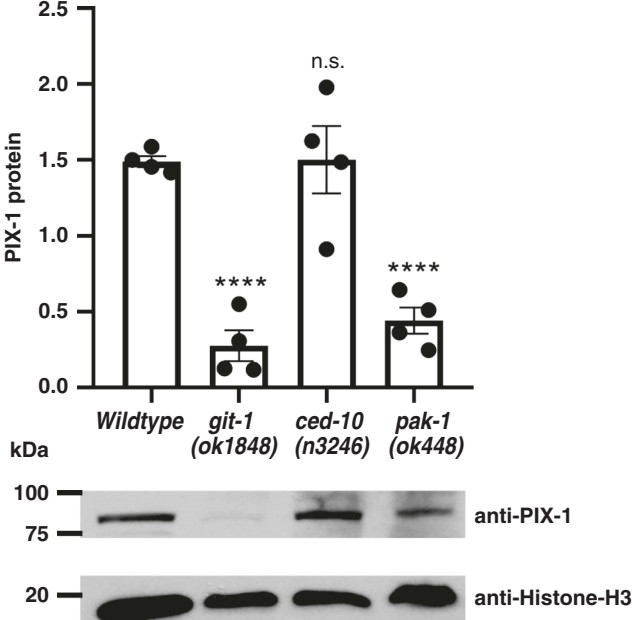

**Fig. 8 PIX-1 levels are reduced in *git-1* and *pak-1* mutants.** Equal quantities of total SDS-soluble proteins from wild type, *git-1(ok1848)*, *ced-10(n3246)*, and *pak-1(ok448)*, were resolved on a gel, blotted to the membrane and reacted with anti-PIX-1, and as a loading control with anti-histone H3. After normalization to the amount of histone H3, the levels of PIX-1 in the mutants were compared to the level of PIX-1 in wild type. Representative immunoblot results are shown below the graph. $N = 4$ independent western blot reactions from each strain; means and standard error of the means are shown; *git-1* and *pak-1* mutants show statistically different levels from wild-type based on a two-sided student's *t*-test at $p \leq 0.0001$ (indicated by ****). Images of the entire western blots are provided in the Source Data Files.

simulations to predict structural effects of the P190S mutation. In PIX-1 simulations, we observed that the serine eliminates the helical kink at the P190 position (Fig. 10a, left pair). This same effect is not observed in Rac1 complexes (Fig. 10a, right pair). The presence of the GTPase appears to maintain the kinked conformation at the 190 position, even with the serine substitution.

However, further investigation of the complexes in MD simulations reveals that the mutation modulates the interaction of PIX-1 with Rac1 in complexes. We evaluated the (i) contact surface area, (ii) van der Waals interactions at the interface, and (iii) root mean square fluctuations (RMSF) of protein residues. Our analyses reveal that the introduction of the P190S mutation stabilizes a putative complex between PIX-1 and GTP-bound Rac1. We observe an increase in the contact surface area between the two proteins in the mutant compared to the wild-type PIX-1 complex (Fig. 10b). We examined a predicted van der Waals interaction on the interface between A186 (PIX-1) and L70 (located in the Switch II region of Rac1). This interaction, which lies in the vicinity of P190, exists in the mutant P190S complex but not in the wild-type complex. In addition, RMSF analysis reveals that the interface is more stabilized in the P190S complex (Supplementary Fig. 6). RMSF values are lower around the interface of the mutant complex, indicative of reduced fluctuations and a more stable interaction.

All the trends that are observed here are reversed in the PIX-1 complexes with GDP-bound Rac1. Contact surface area between PIX-1 and Rac1 is decreased in the mutant complex relative to wild type (Fig. 10b). Consistent with this observation, the van der Waals interaction shown in Fig. 10c is lost in the mutant

complex, but present 24.4% of the simulation in wild type. Combined, MD analyses suggest that the P190S mutation enhances GTP-bound Rac1 interaction with PIX-1.

**P190S reduces the level of activated CED-10 in muscle.** To understand how the P190S mutation results in a normal level of PIX-1 protein but disrupted muscle cell boundary structures, *pix-1(gk893650)* [P190S] was crossed into the transgenic line in which HA-tagged CED-10 is expressed in body wall muscle cells, made whole worm lysates and used GST-PAK-PBD to pull down GTP-bound CED-10. As shown in Fig. 10d, less activated CED-10 was pulled out from *pix-1(gk893650)* [P190S] than from wild type. Repeating this experiment three times, the mean level of activated CED-10 from P190S was 52.9 +/− 5.7% (mean and standard deviation) of the level from wild type. Therefore, the P190S mutation in the RhoGEF domain of PIX-1 reduces its GEF activity.

## Discussion

By screening a collection of adult-viable *C. elegans* mutants by immunostaining, we identified a strain in which IAC components are missing from the muscle cell boundaries but present and normally localized at M-lines and dense bodies. These boundaries consist of cell to ECM to cell attachments. The defect in the strain was mapped to a single mutant gene, *pix-1*, which encodes a PIX protein, known from previous studies to be a RhoGEF for Rac/Cdc42. As compared to wild type, a *pix-1* null mutant shows an ~50% reduction in the level of activated (GTP bound) Rac in muscle. Despite having normally organized sarcomeres, multiple *pix-1* mutants display reduced whole-animal locomotion. We hypothesize that this reduced motility results from decreased transmission of lateral forces between muscle cells. Interestingly, in addition to deficiency of PIX-1, muscle-specific overexpression of PIX-1 protein also results in decreased locomotion and disrupted muscle cell boundaries. Perhaps these results reflect the requirement for PIX-1 signaling to be set at just the optimal level for proper assembly or maintenance of IACs at the muscle cell boundary.

Antibodies to PIX-1 localize to all 3 IACs–muscle cell boundaries, M-lines and dense bodies—and yet PIX-1 is only required at muscle cell boundaries. One possibility is genetic redundancy, that is, there is a second PIX protein that is localized to M-lines and dense bodies that compensates for loss of PIX-1 at these sites. This does not seem to be the case however, since no PIX-1 paralogs can be found by querying the *C. elegans* proteome. Another possibility is that there are additional RhoGEF-containing proteins with RacGEF activity like PIX-1 that are present at M-lines and dense bodies, but not at muscle cell boundaries. We find that there are a total of 17 proteins in *C. elegans* that contain RhoGEF (DH) domains and are expressed in muscle, based on the query of SAGE data[17] (Supplementary Table 1). These 17 proteins include PIX-1, and UIG-1 and UNC-89 that have been studied previously[20,21]. UNC-89 is specifically localized to M-lines[22,23], and the DH domain of UNC-89 specifically activates RHO-1(RhoA)[24], and is thus not relevant here. The properties of UIG-1 somewhat support this hypothesis: UIG-1 is localized to dense bodies and is a GEF for Cdc42. TIAM-1 is reported on WormBase to activate Rac. UNC-73 has two RhoGEF domains, one that activates Rac and one that activates RhoA[16]. The muscle intracellular locations of TIAM-1 and UNC-73 are unknown.

Another possible reason that PIX-1 is required at muscle cell boundaries but not at M-lines dense bodies can be envisioned. We observe the loss of IACs at muscle cell boundary when either the PIX-1 pathway is reduced or increased in activity; there are

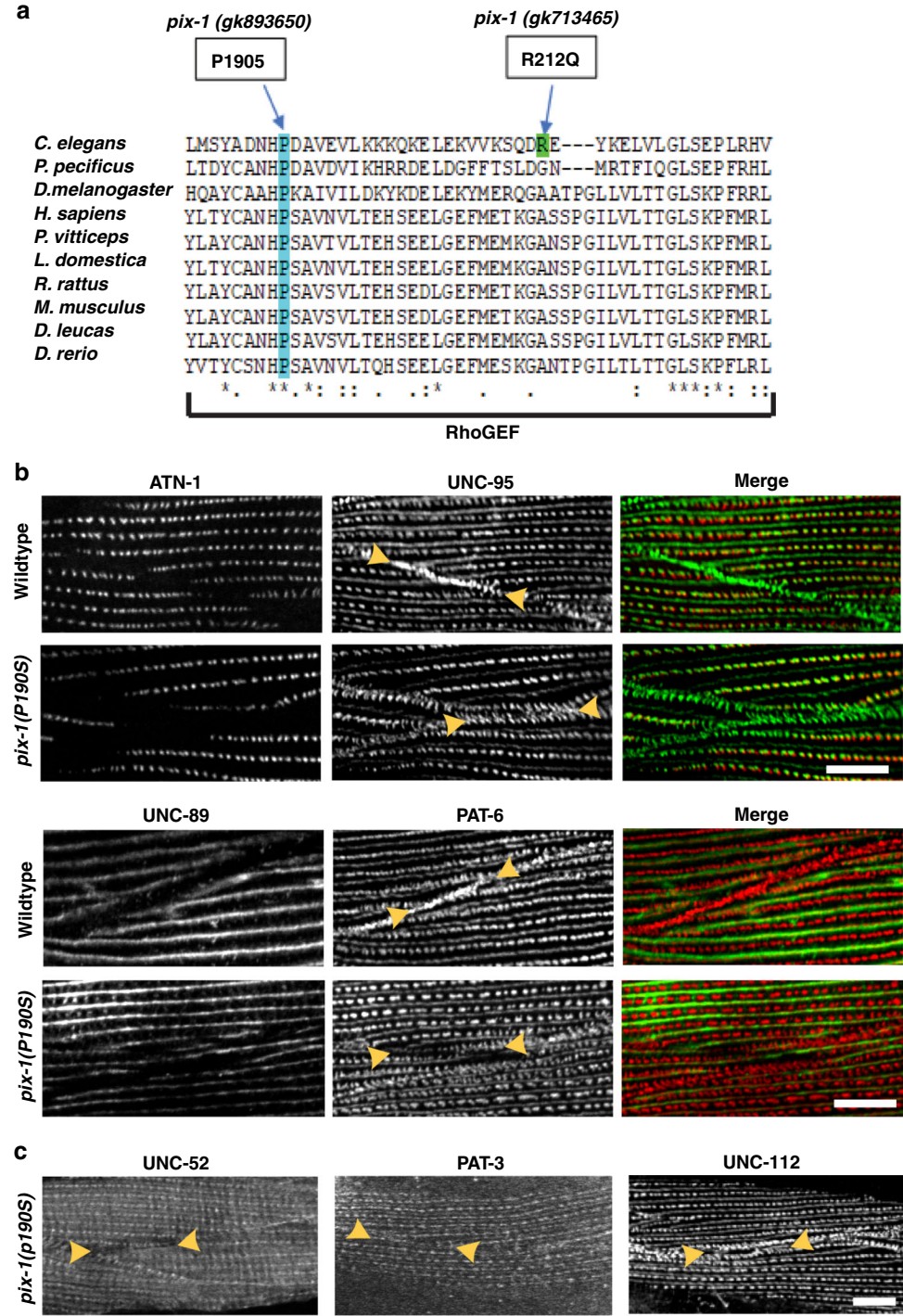

**Fig. 9 P190 is conserved in PIX RhoGEF domains and required for PIX-1 function at muscle cell boundaries. a** PFAM alignment of RhoGEF domain sequences of PIX proteins from 10 species showing that P190 is absolutely conserved. In contrast, R212 is not conserved, which might explain why *pix-1 (gk713465)* [R212Q] has no obvious phenotype. **b** Confocal microscopy of wild type and *pix-1(gk893650)* [P190S] mutant co-stained with antibodies to ATN-1 (α-actinin) and UNC-95 (top two rows), and co-stained with UNC-89 (obscurin) and PAT-6 (α-parvin) (bottom two rows). Note that in *pix-1 (gk893650)*, at the muscle cell boundary, the two halves of the zipper are separated, whereas in wild type they are together. **c** Confocal imaging of *pix-1 (gk893650)* [P190S] stained with antibodies to UNC-52 (perlecan), PAT-3(β-integrin) or UNC-112 (kindlin). Note that these IAC proteins also show reduced or disrupted localization at muscle cell boundaries. Arrowheads bracket a muscle cell boundary. Each image is a representative image obtained from at least 2 fixation and immunostaining experiments, and imaging at least three different animals of each strain. Scale bar, 10 μm.

boundary defects in loss of function for *pix-1, git-1, ced-10,* and *pak-1*, and from overexpression of *pix-1*. These results could be interpreted as indicating a requirement for cycling of the RacGTPase rather than its absolute activity. If IAC assembly required the PIX-1 pathway, then IACs that are assembling and dis-assembling at faster rates would be more sensitive to PIX-1 cycling. Therefore, we have considered the hypothesis that PIX-1 is required at MCBs because muscle cell boundaries are more dynamic that M-lines and dense bodies. However, our preliminary FRAP experiments with GFP tagged PAT-6 and

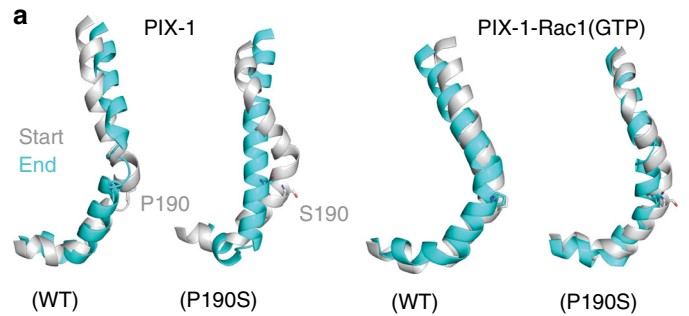

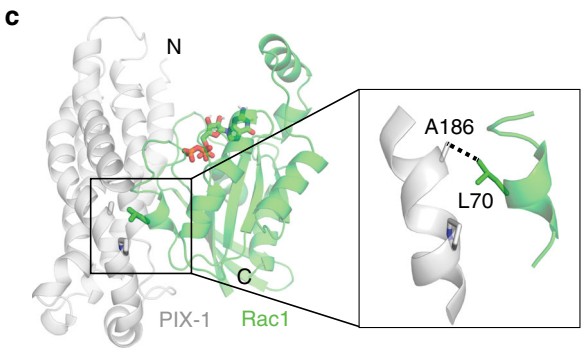

| | Contact area (Å²) |
|---|---|
| WT-Rac1 (GTP) | 1298.3 |
| P190S-Rac1 (GTP) | 1742.0 |
| WT-Rac1 (GDP) | 2047.2 |
| P190S-Rac1 (GDP) | 1597.4 |

| | Distance (Å) | Interaction % |
|---|---|---|
| WT-Rac1 (GTP) | 9.8 | 1.2 |
| P190S-Rac1 (GTP) | 4.4 | 65.5 |
| WT-Rac1 (GDP) | 5.1 | 24.4 |
| P190S-Rac1 (GDP) | 9.1 | 0.4 |

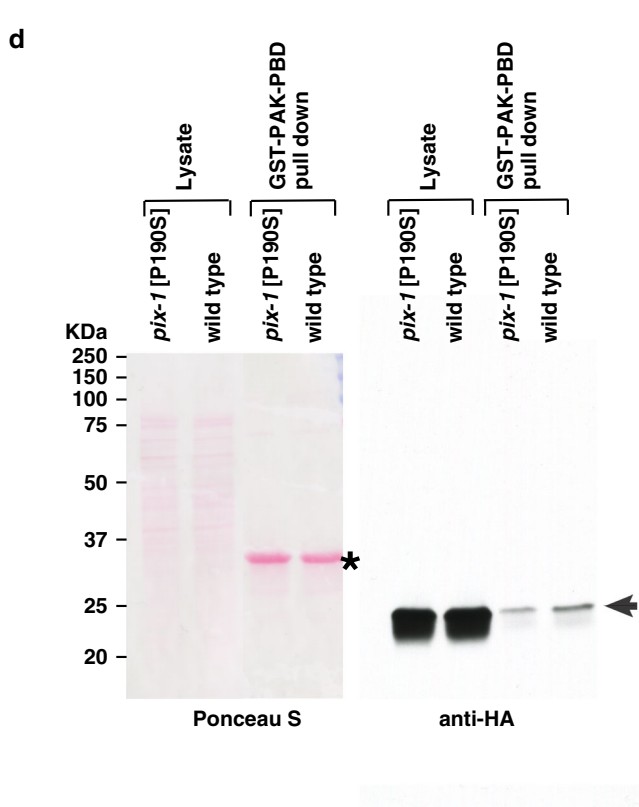

UNC-97, show no differences in turnover rates at muscle cell boundaries vs. dense bodies, so this idea does not seem viable.

It should be emphasized that prior to our study, there were no known genes that when mutated resulted in loss or disorganization of IACs specifically at muscle cell boundaries. This easily scorable phenotype allowed us to determine that each component of the known PIX-1 signaling pathway is required for the assembly or stability of IACs at muscle cell boundaries. Although

*C. elegans* has a single PIX protein, in mammals, there are two PIX proteins, α-PIX and β-PIX, and these have been shown to be important for development and function of nervous and immune systems[25,26]. In *C. elegans*, PIX-1 has been shown to be required for several events during development: control distal tip cell shape and migration (important for germ cell formation)[15], migration of Q neuroblasts that give rise to sensory and inter-neurons[27], tension-dependent morphogenesis of epidermal cells[28], and for

**Fig. 10 P190S may alter RhoGEF structure and interaction with Rac, and in muscle there is a reduction in activated Rac. a** Comparison of helical conformations at the start (gray) and end (cyan) of PIX-1 and PIX-1-Rac1 complex MD simulations. Wild type PIX-1 maintains the predicted kinked helical conformation near P190. (This proline is on helix F in the β-PIX structure.) Mutant PIX-1 reveals a loss of the kink in the helix by the end of the simulation. In simulations of PIX-1 complexed with GTP-bound Rac1, the kinked conformation is maintained in the mutant complex. The same result is observed with GDP-bound Rac1 (not shown). **b** Contact surface areas were calculated for wild type and mutant PIX-1-Rac1 complexes. In GTP-bound Rac1 complexes, the mutation leads to an increase in contact surface area. In GDP-bound Rac1 complexes, the mutation leads to a decrease in the contact surface area. **c** The predicted PIX-1-Rac1 complex reveals an interaction on the interface near the P190 position. This interaction between A186 of PIX-1 with L70 of Rac1 (Switch II region) is quantified as an average distance between the residues and percentage of time the residue distance is <4.5 Å over the simulation. In GTP-bound Rac1, the interaction is lost in wild type but maintained in the mutant. The trend is reversed in GDP-bound Rac1. **d** Comparison of the level of activated CED-10 (Rac) in wild type vs. P190S mutant muscle. Nematode lysates were prepared from two strains, each expressing HA-CED-10 in body wall muscle: wild type, and *pix-1(gk893650)* which expresses approximately normal levels of P190S mutant PIX-1 protein. The lysates were incubated with beads coupled to GST-PAK-PBD to pull down activated CED-10. The Ponceau S stained blot, and the result of the western using anti-HA are shown. Asterisk indicates the position of GST-PAK-PBD on the blot. Arrow indicates the position of HA-CED-10 from the lysate, or HA-CED-10•GTP from the pulldown.

proper early embryonic elongation[29]. However, until our results, no study had demonstrated a function for a PIX protein in striated muscle, in any organism.

By quantitative western blot, we showed that the level of PIX-1 is reduced in loss of function mutants for the scaffold protein GIT-1 and the effector protein kinase, PAK-1, but not for loss function for the Rac protein CED-10. These data suggest that GIT-1 and PAK-1 are required for the stabilization of PIX-1. The GIT-1 result is consistent with the known high-affinity complex formed between PIX and GIT proteins in mammals[30]. It is also consistent with the reports that GIT1 deficient mice have reduced levels of α-PIX and β-PIX in the brain[31], and that GIT2 deficient mice have reduced levels of α-PIX in immune cells[32]. Our finding that PIX-1 is reduced in a *pak-1* mutant is consistent with our finding that by yeast 2 hybrid assays, PIX-1 interacts with PAK-1 (see Supplementary Fig. 7). It is also compatible with the known interaction of the SH3 domain of β-PIX with PAK1-3 in mammals[14].

We have also shown that deficiency of either *pak-1* or *pak-2* results in the muscle cell boundary defect. In the future, we would like to determine if protein kinase activity of PAK-1 or PAK-2 is required for this function, and if so, to identify the key substrates in muscle. Zhang et al.[28] have demonstrated that in *C. elegans*, PAK-1 phosphorylates several intermediate filament proteins in epidermal cells and this is required for maturation of hemidesmosomes in these cells. However, *C. elegans* muscle does not seem to express intermediate filament proteins, and this suggests the existence of additional PAK substrates, at least in muscle.

We were able to find that the MMP collection contains a most informative *pix-1* mutant allele, P190S. This mutation results in a stable full-length PIX-1 protein that localizes in the vicinity of muscle cell boundaries, but yet the boundaries are not properly formed. These data together with the absolute conservation of proline at this position in the RhoGEF domain of PIX proteins, suggest that the GEF activity of PIX-1 is crucial for its function. By expressing CED-10 (Rac) specifically in muscle, we were able to show that using a pull-down assay, worms carrying this PIX-1 P190S mutation have a ~50% reduction in the level of activated (GTP bound) CED-10(Rac) in muscle. This was approximately the same level of reduction in activated CED-10 that we observed in a *pix-1* null mutant. This suggests that, indeed, the P190S mutation completely inactivates the RacGEF activity of PIX-1. In the future, biochemical GEF assays using purified proteins will be necessary to determine if this prediction is true. Nevertheless, the muscle cell boundary defect of *pix-1* null and P190S mutants are different; whereas the null mutants result in the absence of IACs, the P190S mutant results in a defect in which components of the IAC localize to muscle cell boundaries, but the boundary appears split, such that the IACs are formed but are less functional, perhaps less able to anchor cells to the ECM. The fact that the

P190S mutant shows some IAC formation and an abundant full-length protein, and is still likely to have zero RacGEF activity, suggests that perhaps PIX-1 has a function in addition to its RacGEF activity.

Finally, our analysis of a homology model of the RhoGEF domain of PIX-1 bound to Rac1 reveals that the P190S mutation likely increases the affinity of GTP-bound Rac1 with PIX-1. We suggest that this enhanced interaction will slow the release of GTP-bound Rac from the RhoGEF of PIX-1, consequently reduce the probability of GDP-bound Rac from binding to PIX-1, reduce the rate of the GTPase cycle, and thus result in less IAC formation. Again, in the future, we hope to conduct biochemical assays to test this prediction.

## Methods

**C. elegans strains**. All nematode strains were grown on NGM plates using standard methods and maintained at 20° unless otherwise noted. The following mutant and transgenic strains were used: N2 (wild type, Bristol)[33], VC20386 [*pix-1 (gk299374)*], GB291 [*pix-1(gk299374)*; outcrossed 5X to N2], VC863 [*pix-1(gk416)*], GB292 [*pix-1(gk416)*; outcrossed 5X to N2], VC651 [*pix-1(ok982)*], GB294 [*pix-1 (ok982)*; outcrossed 3X to N2], VC30034 [*pix-1(gk406361)*], VC30094 [*pix-1 (gk299384)*], VC40598 [*pix-1(gk713465)*], VC40945 [*pix-1(gk893650)*], GB286 [*pix-1(gk299374)*; *sfEx61[myo-3p::HA::PIX-1a; sur-5::NLS::GFP]*, GB288 [*pix-1 (gk299374)*; *sfIs20[myo-3p::HA::PIX-1a; sur-5::NLS::GFP]*, GB290 [*pix-1(gk299374)*; *sfIs20[myo-3p::HA::PIX-1a; sur-5::NLS::GFP*; outcrossed 1X to *pix-1(gk299374)*], RB689 [*pak-1(ok448)*], RB1540 [*git-1(ok1848)*], MT9958 [*ced-10(n3246)*; received as outcrossed 4X to N2], VC259 [*pak-2(ok332)*; temperature sensitive (ts)], NG103 [*mig-2(gm103)*; received as outcrossed 2X to N2], VC126 [*rac-2(ok326)*], and VC1462 [*max-2(ok1904)*; received as outcrossed 1X to N2]. The ts allele *pak-2 (ok332)* was grown at 15°, embryos prepared, allowed to hatch to L1, and then shifted to 25° for growth to adulthood.

WormBase notes that *git-1(ok1848)* is an ~1 kb deletion. By PCR and sequencing, we determined that *git-1(ok1848)* is a 891 bp deletion that begins after the codon for amino acid 550 and continues into the 3'UTR, where, after an additional 29 codons occur in-frame a premature stop codon is encountered. This results in a mutant GIT-1 protein that is the missing the normal C-terminal 120 residues (normally GIT-1 is 670 residues long), but has 29 residues of novel sequence at its C-terminus.

**CRISPR/Cas9 correction of pix-1(gk299374) to wild-type sequence**. The conversion of the 153$^{rd}$ codon of *pix-1* from the TAA stop codon in GB291, *pix-1 (gk299374)* 5X o.c., to a CAA Gln codon (as found in wild-type strain N2), was conducted by SunyBiotech (http://www.sunybiotech.com) using the following sgRNA and repair template to generate strain PHX2137, *pix-1(syb2137gk299374)*: sgRNA, AACTGGAGCGTGAGCAAAAg(tga) (parentheses bracket the "NGA" site); repair template: ACGTTGGTTGGGAATTTCGAAGTAATTTACACTCTGAAACGTGA TCTTTTTGAGCAATTGGAGCGCGAGCAAAAgtgagttttaatttctccaaccttcttcaactttctt atttcagTGAG

Since there were no suitable "NGG" sequences close to the sgRNA for the editing site, an "NGA" was used together with mutant Cas9 (D1135V R1335Q T1337R). PHX2137 was then outcrossed 3X to generate strain GB317, which was used for locomotion assays.

**Immunostaining of body wall muscle**. Adult nematodes were fixed and immunostained using the method described by Nonet et al.[34] with further details provided in Wilson et al.[35]. In brief, synchronized adults were washed free from *E. coli* by multiple washes in M9 buffer and ~50 μl of packed worms were fixed with

810 µl of fixative (50% Bouin's fixative, 50% methanol, 1.2% β-mercaptoethanol) at room temperature for 30 min, then frozen in liquid nitrogen for 5 min, thawed and continuing incubation at room temperature for an additional 30 mins. The worms were pelleted and washed 3× with 1.4 ml of BTB solution (20 mM sodium borate pH 9.5, 0.5% Triton X-100, 2% β-mercaptoethanol), and then resuspended in 1 ml of BTB and continuing incubation with mixing for 1 h. The worms were pelleted again and resuspended in 1 ml of BTB and incubated for 3 hrs with mixing at room temperature. The worms were pelleted and washed with 1 ml of BT (20 mM sodium borate pH 9.5, 0.5% Triton X-100), pelleted, and washed 2× with 1 ml of AbA buffer (PBS, 1% bovine serum albumin, 0.5% Triton X-100, 1 mM sodium azide, 1 mM EDTA), pelleted, resuspended in 1 ml of AbA and incubated with mixing for 30 mins, pelleted and resuspended in 100 µl of AbA. Immunostaining was conducted using 5 µl of a suspension of these fixed worms together with 20 µl of primary antibody in AbA and incubating overnight with mixing. The animals were pelleted and washed 4X with PBS + 0.5% Triton X-100, pelleted, removing as much supernatant as possible and incubating with 20 µl of secondary antibody in AbA for 2 h, followed by washing 4× with PBS + 0.5% Triton X-100, and finally removing as much supernatant as possible and mounting 5 µl of resuspended worms with 5 µl of DAPCO solution (20 mM Tris-HCl pH 8.0, 0.2 M 1,4-diaza-bicyclo-2,2,2-octane (DABCO), 90% glycerol) on a glass slide with a coverslip and sealed with nail polish. The following primary antibodies were used at 1:200 dilution except as noted: anti-PAT-6 (rat polyclonal)[36], anti-UNC-52 (mouse monoclonal MH2)[37], anti-PAT-3 (1:100 dilution; mouse monoclonal MH25)[38,39], anti-UNC-95 (rabbit polyclonal Benian-13)[40], anti-UNC-112 (1:100 dilution)[20], anti-MHC A (mouse monoclonal 5–6)[41], anti-UNC-89(rabbit polyclonal EU30)[22], anti-ATN-1 (mouse monoclonal MH35)[42], anti-HA (mouse monoclonal; H3663; Sigma-Aldrich), and anti-GFP (rabbit polyclonal; Thermo Fisher, A11122). Secondary antibodies, used at 1:200 dilution, included anti-rabbit Alexa 488, anti-rat Alexa 594, and anti-mouse Alexa 594, all purchased from Invitrogen. Fixation and phalloidin-rhodamine staining staining was conducted as described[43]. Images were captured at room temperature with a Zeiss confocal system (LSM510) equipped with an Axiovert 100 M microscope and a Apochromat x63/1.4 numerical aperture oil immersion objective, in 1× and 2.5× zoom mode. For the images presented in Fig. 2, and Supplemental Figs. 2 and 5, super-resolution microscopy was performed with a Nikon N-SIM system in 3D structured illumination mode on an Eclipse Ti-E microscope equipped with a 100×/1.49 NA oil immersion objective, 488- and 561-nm solid-state lasers, and an EM-CCD camera (DU-897, Andor Technology). Super-resolution images were reconstructed using the N-SIM module in NIS-Elements software. For all the images, confocal, and SIM, the color balances were adjusted by using Adobe Photoshop (Adobe, San Jose, CA).

**Mapping the phenotype to *pix-1*.** The original MMP strain VC20386[12] has outcrossed to wild type N2 Bristol three times, each time selecting for the PAT-6 muscle boundary defect. Comparison of genomic sequences of VC20386 and N2 allowed the selection of SNPs on each chromosome arm to distinguish the two strains. PCR and Sanger sequencing of these 12 segments from the third outcrossed strain revealed that only the left arm of chromosome III and the right arm of X were derived from VC20386. For III and X, we identified genes known to be expressed in muscle[17], and of these identified four genes on III and three genes on X, that had nonsense or non-conservative missense mutations in VC20386. RNAi for three of them failed to reveal the PAT-6 boundary defect. One of the 7 genes, *pix-1* on X, had a nonsense mutation. Six additional mutant alleles of *pix-1* were obtained from CGC, and four of these six mutants also showed the PAT-6 boundary defect (detailed in Results).

**Transgenic rescue.** An HA-tagged cDNA encoding full-length PIX-1a was amplified by PCR using the RB2 cDNA library (provided by Robert Barstead) as template and was cloned into vector pPD95.86 (provided by Andrew Fire) designed to express HA-PIX-1a in muscle by the *myo-3* muscle-specific promoter. This plasmid, pPD95.86-HA-PIX-1a at 10 ng/µl together with plasmid pTG96 which expresses SUR-5-NLS-GFP as a transformation marker at 90 ng/µl was microinjected into *pix-1(gk299374)* 5X o.c. to generate the strain GB286, *pix-1(gk299374); sfEx61*[*myo-3p*::HA::PIX-1a; *sur-5*::NLS::GFP]. This extrachromosomal array was integrated into the genome by ultraviolet irradiation[44] with modifications (Peter Barrett, personal communication). The resulting nematode strain is called GB288, *pix-1(gk299374); sfIs20* [*myo-3p*::HA::PIX-1a; *sur-5*::NLS::GFP]. To remove background mutations induced by UV, GB288 was backcrossed to *pix-1(gk299374)* 1X, recovering strain GB290, and this strain was tested in locomotion assays. The extrachromosomal array *sfEx61* [*myo-3p*::HA::PIX-1a; *sur-5*::NLS::GFP], was also crossed into N2 to generate the strain GB295.

**Imaging of F-actin at the muscle cell boundary.** We used strain KAG547 kindly provided by Kathrin Gieseler (Universite Claude Bernard Lyon): GFP::MYO-3; *Ex [myo-3p::LifeAct::mCherry]*. Animals were washed off of plates and washed free from bacteria using M9 buffer and then immobilized by incubation in 10 µM levamisole in M9 for 10 min. Approximately 50–100 animals in 3 µl were added to 7 µl of ice-cold 25% Pluronic F127 in M9[45] lying on a cold glass slide, to which was added a cover slip and it was sealed with nail polish. After incubation at room

temperature for 5–10 min to solidify, images were taken using a Nikon N-SIM microscope system, as described above.

**Protein sequence analysis.** Nematode PIX-1a protein sequence was obtained from Wormbase. A BLAST homology search identified human orthologs of the nematode protein using the NCBI PubMed database. The domain organization for PIX-1 and its orthologs was analyzed via PFAM (Fig. 3a). Human α-PIX and β-PIX amino acid sequences were aligned with PIX-1 using pBLAST to determine percent identities for each domain (Fig. 3a). PIX-1 RhoGEF domains from 10 organisms were compared also using pBLAST (Fig. 9a).

**Measurement of CED-10 activation state in body wall muscle.** An HA-tagged cDNA encoding full-length CED-10 was amplified by PCR using the RB2 cDNA library as template and was cloned into vector pKS-HA8(Nhex2), the DNA sequenc- verified, and then the NheI fragment was excised and inserted into pPD95.86 designed to express HA-CED-10 in muscle from the *myo-3* muscle-specific promoter. This plasmid, pPD95.86-HA-CED-10, at 10 ng/µl together with plasmid pTG96 which expresses SUR-5-NLS-GFP as a transformation marker at 90 ng/µl was microinjected into wild-type animals to generate a transgenic strain GB314, *sfEx63*[*myo-3p*::HA::CED-10; *sur-5*::NLS::GFP]. This extrachromosomal array was integrated into the genome by ultraviolet irradiation[44] with modifications (Peter Barrett, personal communication). The resulting nematode strain is called GB315, *sfIs22* [*myo-3p*::HA::CED-10; *sur-5*::NLS::GFP]. This strain was crossed into *pix-1(gk299374)* 5X OC, to generate strain GB316, *pix-1(gk299374); sfIs22*, and crossed into *pix-1(gk893650)* [P190S] 5X OC, to generate strain GB318, *pix-1 (gk893650); sfIs22*. After growing several grams of worms from GB315, GB316, and GB318, worm powders were prepared by grinding in a mortar and pestle under liquid nitrogen on a bed of dry ice. We modified the Rac1 Activation Assay Biochem Kit (cat. #BK035, Cytoskeleton, Inc.) as follows: One small spatula-full of worm powder was added to 3 ml of ice-cold Cell Lysis Buffer containing protease inhibitor cocktail, vortexing for 1 min, and centrifuging at maximum speed in a microcentrifuge for 10 min at 4°. A small portion supernatant was used for protein concentration determination using the BCA Assay (cat. #23225, ThermoScientific), and multiple 200 µl aliquots of the remainder were snap frozen in liquid nitrogen and stored at −70°. The protein concentrations of the lysates varied from 1.25 to 3.4 mg/ml. Positive and negative controls were created by adding to 250 µg of total protein of wild-type worm lysate, GTPγS to a final concentration of 0.20 mM, or GDP to a final concentration of 1 mM, respectively. 250 µg of total protein from each lysate were added to 10 µg of GST-PAK-PBD Beads, and incubated with mixing for 1 h, 15 min, at 4°. Then, the beads were pelleted at 4° by spinning at $4000 \times g$ for 1 min, supernatant carefully removed, and the beads were washed 1× with Wash Buffer, and the beads pelleted. After removing as much supernatant as possible, 20 µl of 2× Laemmli were added, vortexed for 5 s, heated at 95° for 3 min, centrifuged at top speed for 3 min, and then 20 µl of each supernatant were run on a 12% SDS-PAGE and transferred to nitrocellulose. HA-CED-10 was visualized by incubating with rabbit monoclonal antibodies to HA at 1:1000 dilution (cat. #C29F4, Cell Signaling Technology), and reacting with ECL reagents and exposed to film. We used a flat-bed scanner to image the Ponceau S staining of the blot using the "reflective" mode in 24 bit color, and to image the ECL reactions recorded on film using the "film scan" mode in gray scale. The images were opened in AdobePhotoshop, and after inverting the ECL images, both the ECL bands (HA-CED-10) and the Ponceau S bands (GST-PAK-PBD) "mean" and "pixel" values were recorded. The absolute intensity of each band was a product of these two values. The HA-CED-10 band products were normalized by dividing by the GST-PAK-PBD bands for each lane.

**Swimming and crawling assays.** For swimming assays, day 2 adults from two, 6 cm NGM OP50 seeded plates were washed off the plates, washed free from bacteria and collected into M9 buffer such that the ratio of worms to buffer was 1:1. 2 ml of M9 buffer was added to an unseeded 6 cm NGM plate where upon 5 µl of worm suspension was added to the center of the plate. Worms were allowed 5 min to adapt before a video recording of their swimming motions was made using a dissecting stereoscopic microscope fitted with a CMOS camera (Thorlabs). Ten, 10 s videos were recorded for each nematode strain from different sections of the plate, each video tracking the motion of ~10 worms. The video data were analyzed by Image J FIJI WrmTracker software[46] to obtain body bends per second (BBPS). After removing outliers and animals that had moved out of frame, ~30 animals were analyzed for each strain. The resulting BBPS values for each mutant strain were compared to wild type and differences were tested for statistical significance using a Student *T*-test.

For crawling assays, day 2 adults were harvested as above, except that all washing steps used M9 buffer containing 0.2 g/L gelatin. Five microliters of worm suspension was added to the center of a 6 cm unseeded NGM plate, and the excess liquid was removed. After 5 min for adaptation, worm crawling was recorded using the above-mentioned strategy for extraction of BBPS for individual worms in each video. The resulting values for each strain were compared to wild type for statistical analysis using a Student *T*-test for significance.

**PIX-1 antibody generation**. Rabbit polyclonal antibodies were generated to two different immunogens from PIX-1a. Immunogen #1 consists of residues 268–449, and immunogen #2 consists of residues 441–646. The coding sequences for each immunogen were cloned into vectors pGEX-KK1 and pMAL-KK1, respectively to express glutathione S-transferase (GST) and MBP fusion proteins in *E. coli*[47]. In brief, to express each fusion protein, the plasmids were transformed into *Esherichia coli* Rosetta 2 (DE3) (Millipore Corporation, cat. no. 71397-4), and grown in Luria-Bertani medium containing 10 µg/ml ampicillin at 37° to $OD_{600} = 0.6–0.8$, followed by induction of protein expression with 0.5 mM isopropyl β-D-1-thiogalactopyranoside, and further growth at 20° for 5 h. Cell pellets were harvested by centrifugation and resuspended in 50 mM Tris-HCl pH 8.0 supplemented with 100 mM phenylmethylsulfonyl fluoride (PMSF) and cOmplete Mini protease inhibitor cocktail (Roche, Inc., cat. no. 11836170001), and then broken in a French pressure cell at 1000 pounds inch$^{-2}$, addition of Triton X-100 to 1%, followed by spinning out debris by centrifugation at $12,000 \times g$ for 20 min. The cleared lysates were added to either a slurry of glutathione-agarose beads (Sigma, cat. no. G-4510) or amylose resin (New England BioLabs, cat. no. E8021L), and incubated with shaking for 30 min at 4°. The beads were then washed 5× with 50 mM Tris-HCl pH 8.0 containing 1% Triton X-100 and 100 mM PMSF, and then 4× with 50 mM Tris-HCl pH 8.0 containing 100 mM PMSF. The beads/resin were placed onto mini columns and then eluted with either 10 mM free glutathione or 10 mM maltose in 50 mM Tris-HCl pH 8.0. The GST fusions were shipped to Noble Life Sciences (Sykesville, Maryland) for the production of rabbit antibodies. Antibodies were affinity-purified using Affigel (BioRad)-conjugated MBP fusions, as described[48]. In brief, the MBP fusion proteins were dialyzed against 100 mM 3-morpholinopropane-1-sulfonic acid (MOPS) pH 7.0, and then 3–10 mg were covalently coupled to ~1 ml of a 50:50 mixture of Affi-Gel 10 and Affi-Gel 15 beads (Bio-Rad Laboratories, cat. nos. 1536099 and 153-6051) according to the manufacturer's procedure. These columns were then used to affinity purify ~2 ml of antiserum and the antibodies eluted with low and high pH and concentrated with a centrifugal filter (Centriprep 10, Millipore, cat. no. 4304). The method of Hannak et al.[49] was used to prepare total protein lysates from wild-type, 5X *pix-1(gk416)* o.c., *pix-1(gk982)*, *pix-1(gk299374)* 5X o.c., *pix-1(gk893650)*, *git-1(ok1848)*, *ced-10(n3246)*, and *pak-1(ok448)* mixed-stage animals. In brief, we grew worms on 2–4, 10 cm NGM plates seeded with *E. coli* OP50, washed the worms off with M9 buffer and continued washing until the supernatant was clear of bacteria, yielding ~100 µl of worms after centrifuging briefly in a microfuge. To make protein lysates, an equal volume of 2× Laemmli sample buffer containing EDTA and a protease inhibitor cocktail was added, vortexed 1 min, and sonicated for 10 mins in a water bath sonicator containing 80 °C water. After a quick pop spin, the material was heated in a boiling water bath for 5 min, and then debris was pelleted by spinning for 5 min at top speed in a microfuge. Equal amounts of total protein were separated on 10% polyacrylamide-SDS- Laemmli gels, transferred to nitrocellulose membranes, reacted with affinity purified, *E. coli*-OP50-absorbed anti-PIX-1a at 1:200 (for immunogen #1) or at 1:5000 (for immunogen #2), reacted with goat anti-rabbit immunoglobulin G conjugated to HRP (GE Healthcare) at 1:10,000 dilution, and visualized by ECL. For total protein loading control, we used reaction to either paramyosin (monoclonal 5–23[41]; 1:5000 dilution), or to histone H3 (rabbit polyclonal ab1791, Abcam, Inc.; 1:40,000 dilution).

**Model construction for molecular dynamics simulations**. Six complexes were prepared for MD simulations. (1) A PIX-1 homology model prepared using PDB 1BY1 (human beta-PIX) as a template[19]. Our model aligned PIX-1 residues 89-272 with the beta-PIX DBL homology domain. (2) A PIX-1-Rac1 complex prepared using PDB 5O33 (GEF Kalirin DH domain in complex with GDP-bound Rac1 GTPase) as a template. (3) PIX-1-Rac1 complex with GTP-bound Rac1, obtained from PDB 6BC1. For consistency, Rac1 sequences were restored to wild type in both GDP- and GTP-bound forms. Complexes 4–6 were P190S mutants of complexes 1–3 respectively. All mutations were introduced in silico.

**Molecular dynamics simulations**. The complexes were solvated in an octahedral box of TIP3P water with a 10 Å buffer around the protein complex. Na+ and Cl− ions were added to neutralize the protein and achieve physiological buffer concentrations. Xleap in AmberTools 18[50] was used to prepare systems for simulation with the parm99-bsc0 forcefield[51]. Parameters for GDP and GTP in Rac1 complexes were created using Antechamber[52] Minimizations and simulations were performed with Amber18[50]. Systems were minimized with 5000 steps of steepest descent followed by 5000 steps of conjugate gradient minimization with 500 kcal/mol Å² restraints on all atoms. In PIX-1-Rac1 complexes, an extra minimization step (5000 steps of steepest descent, 5000 steps of conjugate gradient) was performed, with restraints retained on protein residues lying on the interface between PIX-1 and Rac1. Restraints were then removed from all atoms and both conjugate gradient and steepest descent minimization were repeated.

Following minimization, the systems were heated from 0 to 300 K with a 100-ps run, 5 kcal/mol Å² restraints on all protein/nucleotide atoms and using constant volume periodic boundaries. MD equilibration was performed for 10 ns with 10 kcal/mol Å² restraints on all solute atoms, using the NPT ensemble. Restraints were reduced to 1 kcal/mol Å² which was followed by an additional 10 ns of MD. Restraints were then removed and 500 ns production simulations obtained for each system. All bonds between heavy atoms and hydrogens were fixed with the SHAKE algorithm[53] permitting the use of a 2-fs timestep. Long-range electrostatics and van der Waals forces were calculated with a 10 Å cutoff distance.

*Analysis*. 25,000 evenly spaced frames were obtained from each simulation and used for analysis. Structural averaging and analysis were performed with the CPPTRAJ module of AmberTools[54]. RMSF analysis was performed on Cα atoms of protein residues to calculate atomic deviations over the course of the simulation. RMSF was computed relative to the starting structure. CPPTRAJ was used to calculate distances between heavy atoms of residue pairs over trajectories. Solvent-accessible surface areas (SASA) of proteins were calculated using the molsurf algorithm in AmberTools. Contact surface areas in protein complexes were calculated using the formula $(SASA_{PIX-1} + SASA_{Rac1}) - SASA_{complex}$.

**Reporting summary**. Further information on research design is available in the Nature Research Reporting Summary linked to this article.

## Data availability
The authors declare that the data supporting the results of this study are available within the paper, or available upon request. Source data are available as a Source Data file.

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

## Acknowledgements

We thank SunyBiotech Corporation for generation of the transgenic rescued line GB286, for the CRISPR/Cas9-corrected strain PHX2137, and for the myo-3p::HA::CED-10 transgenic line GB314; Kathrin Gieseler (Universite Claude Bernard Lyon) for nematode strain KAG547, Robert Barstead (Oklahoma Medical Research Foundation) for the cDNA library RB2, Andrew Fire (Stanford University) for plasmid pPD95.86, and Peter Barrett (Xavier University of Louisiana) for his protocol for integrating transgenic arrays. Most of the nematode strains used in this work were provided by the Caenorhabditis Genetics Center, which is funded by the NIH Office of Research Infrastructure Programs (P40 OD010440). This study was supported in-part by a NSF-Graduate Research Fellowship (DGE 1444932) to J.C.M., a NIH grant (R01DK115213-01S) to C.D.O., and from a NIH grant (R01AR064307) to G.M.B.

## Author contributions

J.M. carried out most experiments. A.R.D., C.D.O., N.S., Y.M., and C.J.C. contributed to some experiments. J.M., H.Q., C.D.O., E.A.O., and G.M.B. designed and interpreted experiments and wrote the paper.

## Competing Interests

The authors declare no competing interests
