## [Peer Review File · Nature Communications]

Reviewers' Comments:

Reviewer #1:

Remarks to the Author:

In "The Rho-GEF PIX-1 Directs Assembly of Lateral Attachment Structures Between Muscle Cells" the authors identify a novel regulator of Integrin Attachment Complexes (IACs) in *C. elegans* muscle.

Using mutations from the Million Mutation Project (MMP) the authors identify 5 out of 7 mutations in *pix-1* impact the IACs at muscle cell boundaries, though not immediately obviously at the M-lines or Dense Bodies. As correctly stated by the authors this is the first known mutation to specifically affect the IACs at muscle cell boundaries. Notably, four of the five mutations that impair the muscle cell boundaries disrupt the RhoGEF domain, including a point mutation in an evolutionarily conserved P that the authors show via dynamic structural modelling is required for proper function of the RhoGEF function of PIX-1. The final mutation that impacts the IACs at the muscle cell boundaries is in the coil-coil which suggests that PIX-1 GEF functions as a dimer and/or requires protein-protein interactions for full function, as demonstrated by the authors experiments with *git-1* and *pak-1*. These additional experiments with *git-1*, *ced-10*, *mig-2*, *rac-2*, *max-2*, *pak-1*, and *pak-2* further support the mechanism proposed by the genetic analysis and structural modelling, *in vivo*. Of note the authors experiments with *git-1* further expand our knowledge of how GIT-1 impacts IACs in *C. elegans* muscle to a more mechanistic understanding than the previous RNAi screen results by others and the authors work with *pak-1* and *pix-1* appear to confirm, *in vivo*, the nature of the interaction between these proteins in COS-7 cells. Lastly, the authors confirm that *pix-1* is required for normal muscle function, moreover, the precise concentration of PIX-1 not just any amount of PIX-1 is important (in keeping with both the GEF and coil-coiled data). These very nice experiments elegantly support the authors statements.

From a genetics point of view this work is important as it highlights both how one can isolate and confirm meaningful molecular genetic insight from the MMP and how one can use histologic markers in genetic screening as a starting point rather than as a follow up to gross phenotypic screening. Furthermore, the work is important as it shows for the first time the role of PIX-1 in maintaining muscle homeostasis. This explains what the expected role ARHGEF7, human beta-PIX, is in human muscle, where it is known to be expressed and where at least one SNP in the RhoGEF domain has been reported. Accordingly, it is interesting to note that in Supplementary Figure 2 the *pix-1* mutant appears to have more "wavy" structures as often seen in *C. elegans* DMD mutants.

From an IAC point of view this work is important as it highlights the previously undemonstrated role of PIX-1 in IACs in *C. elegans* muscle and, for the first time, demonstrates a genetic distinction between the muscle cell boundary IACs and the M-lines and Dense Bodies. Thus, this work now enables further mechanistic and phenotypic characterization of PIX-1 *in vivo* and provides the first tool for dissecting what is different about muscle cell boundary IACs from the other muscle IACs. For human muscle the IACs are known to be important in adaptation to both use (exercise) and disuse (cast immobilization, spaceflight) but which subset of IACs is involved remains unclear. For *C. elegans* muscle, this work potentially allows future linking of the various mutations that the authors lab has previously isolated with protein aggregates in the muscle cell boundaries with defects in the muscle cell boundary IACs.

From a muscle point of view this work is incredibly exciting as stated above it potentially links directly to human genetic variation in the same gene in human muscle and to the known role of the IAC in muscle adaptation to mechanical load. In this regard the authors finding of a more pronounced effect of *pix-1* mutation or overexpression on crawling (which is under increased mechanical strain/stress) than with swimming is fascinating and I look forward to future experiments using the *pix-1* mutants with the recently published burrowing assay and strength assay both of which revealed defects in worm DMD mutants (I also look forward to future human experiments looking for alterations in ARHGEF7 with use/disuse). Lastly, I look forward to future

experiments using the *pix-1* mutants vs. the worm DMD mutants (e.g. does *pix-1* also impact mitochondrial structure/function and/or calcium handling like the DMD mutants (alternatively does *pix-1* impact different muscle to muscle channels (for example H channels like in human cardiac muscle) and/or impact pharyngeal muscle (ARHGEF7 has higher expression in human cardiac muscle than in skeletal)).

In sum, this is a paper of broad importance across a number of related yet distinct fields that is comprised of well carried out, well reasoned, and nicely complementary experiments that are clearly of interest and immediate translatability into humans.

Minor point:

The authors provide a lot of data that it is the dynamic nature of the PIX-1 GEF function that contributes to the muscle cell boundary IAC defect in *pix-1* mutants and have been careful to state "is required for assembly or stability of IACs" in the introduction. Thus, could/should stability appear in the title? (I'm not sure as from a biochemical perspective re-assembly is still assembly)

Reviewer #2:

Remarks to the Author:

C. elegans provides a powerful tool to identify and investigate gene products important for cell adhesion and muscle attachment. This work reports the identification of roles for the exchange factor PIX in integrin-mediated adhesion complexes at muscle borders. PIX had already been implicated in adhesion signaling and in regulation of adhesion turnover, although not specifically in muscle so the current work extends observations to now include *C. elegans* muscle. Signaling pathways in which PIX operates have also been reported and similar pathways seem to be at play in the muscle. Thus, while this carefully performed study convincingly shows that PIX-1 is important for nematode muscles the advances in mechanistic understanding are very limited. That PIX-1 impacts a subset of IACs is interesting and surprising but again the basis for this is not explored and the general interest of the message of this report seems small.

Screening of mutant strains reveals a role for PIX-1 in assembly of muscle attachment sites at the muscle boundary and the absence of functional PIX-1 perturbs assembly of these structures, apparently without altering other adhesion structures in the muscle. The ability of PIX-1 re-expression on the mutant background to restore IAC localization strongly supports a role for PIX in this process. The authors infer that this loss of adhesion contributes to defective nematode motility possibly through alterations in force transmission (Fig 3) but here the rescue construct fails to rescue motility (despite apparently rescuing adhesion in Fig 1) and the expression of the rescue construct on the wild-type background also impaired motility – presumably without altering adhesions but this was not assessed. Based on these data it seems difficult to conclude that the PIX effect on adhesions contributes to the motility defect. In this regard it might be interesting to determine whether the PIX mutants that do not alter IACs (Fig 1) exhibit any motility defects.

That PIX-1 localizes to all three IACs in muscle cells seems convincing but the conclusion that the P190S mutant does so is much less convincing from the image shown in Fig 4. Is it possible that it localizes to M lines and dense bodies but not boundaries? Does the 893650 mutant impair localization of integrin, kindlin etc. boundaries (like the deletion mutants do)?

Some of the data on PIX-1 protein levels are difficult to rationalize – while the effect of the scaffolding protein GIT-1 is reasonable it is surprising that PAK1 but not *ced-10* is also required. It would be interesting to probe these results further to attempt to understand why the PAK but not its upstream activator is important.

The MD simulations of the P190S GIT-1 mutant may offer the starting point for further studies but

without a more detailed investigation of the binding interactions or of the GEF activity the modeling predictions have limited utility.

July 27, 2020

Dear Reviewers,

Thank you very much for taking the time and effort to carefully review the previous version of our manuscript, NCOMMS-19-23879-T. We appreciate your generally positive response and constructive suggestions. Based on these suggestions and criticisms, all but one from Reviewer #2, we have substantially revised the manuscript. We apologize for taking such a long time to revise it as we exerted considerable effort into experimentally addressing the concerns of Reviewer #2. Also, we have been impeded from conducting much experimental work since the coronavirus crisis began in March.

Substantially more data has been added. New figures are Figures 2 (live imaging of cortical F-actin at muscle cell boundaries), 4 (comparing the level of activated Rac in muscle in wild type vs. *pix-1* null), 6 (overexpression of PIX-1 results in disrupted muscle cell boundaries), 7 (CRISPR repair of a *pix-1* nonsense mutant results in normal muscle cell boundaries), 13 (comparing the level of activated Rac in muscle in wild type vs. *pix-1* P190S mutant), and a Supplemental Table 1 (RhoGEF domain proteins expressed in *C. elegans* muscle). Additional results have also been added to the old Fig. 3 (now 5), old Fig. 4 (now 8), and to old Fig. 7 (now 11). In the revised manuscript, new text is denoted in blue font.

Below you will see the original and complete comments of the Reviewers in *italics*, and our responses following in non-italic font.

“Reviewer #1 (Remarks to the Author):

In “The Rho-GEF PIX-1 Directs Assembly of Lateral Attachment Structures Between Muscle Cells” the authors identify a novel regulator of Integrin Attachment Complexes (IACs) in C. elegans muscle.

*Using mutations from the Million Mutation Project (MMP) the authors identify 5 out of 7 mutations in *pix-1* impact the IACs at muscle cell boundaries, though not immediately obviously at the M-lines or Dense Bodies. As correctly stated by the authors this is the first known mutation to specifically affect the IACs at muscle cell boundaries. Notably, four of the five mutations that impair the muscle cell boundaries disrupt the RhoGEF domain, including a point mutation in an evolutionarily conserved P that the authors show via dynamic structural modelling is required for proper function of the RhoGEF function of PIX-1. The final mutation that impacts the IACs at the muscle cell boundaries is in the coil-coil which suggests that PIX-1 GEF functions as a dimer and/or requires protein-protein interactions for full function, as demonstrated by the authors experiments with *git-1* and *pak-1*. These additional experiments with *git-1*, *ced-10*, *mig-2*, *rac-2*, *max-2*, *pak-1*, and *pak-2* further support the mechanism proposed by the genetic analysis and structural modelling, in vivo. Of note the authors experiments with *git-1* further expand our knowledge of how GIT-1 impacts IACs in *C. elegans* muscle to a more mechanistic understanding than the previous RNAi screen*

results by others and the authors work with pak-1 and pix-1 appear to confirm, in vivo, the nature of the interaction between these proteins in COS-7 cells. Lastly, the authors confirm that pix-1 is required for normal muscle function, moreover, the precise concentration of PIX-1 not just any amount of PIX-1 is important (in keeping with both the GEF and coil-coiled data). These very nice experiments elegantly support the authors statements.

From a genetics point of view this work is important as it highlights both how one can isolate and confirm meaningful molecular genetic insight from the MMP and how one can use histologic markers in genetic screening as a starting point rather than as a follow up to gross phenotypic screening. Furthermore, the work is important as it shows for the first time the role of PIX-1 in maintaining muscle homeostasis. This explains what the expected role ARHGEF7, human beta-PIX, is in human muscle, where it is known to be expressed and where at least one SNP in the RhoGEF domain has been reported. Accordingly, it is interesting to note that in Supplementary Figure 2 the pix-1 mutant appears to have more “wavy” structures as often seen in C. elegans DMD mutants.

From an IAC point of view this work is important as it highlights the previously undemonstrated role of PIX-1 in IACs in C. elegans muscle and, for the first time, demonstrates a genetic distinction between the muscle cell boundary IACs and the M-lines and Dense Bodies. Thus, this work now enables further mechanistic and phenotypic characterization of PIX-1 in vivo and provides the first tool for dissecting what is different about muscle cell boundary IACs from the other muscle IACs. For human muscle the IACs are known to be important in adaptation to both use (exercise) and disuse (cast immobilization, spaceflight) but which subset of IACs is involved remains unclear. For C. elegans muscle, this work potentially allows future linking of the various mutations that the authors lab has previously isolated with protein aggregates in the muscle cell boundaries with defects in the muscle cell boundary IACs.

From a muscle point of view this work is incredibly exciting as stated above it potentially links directly to human genetic variation in the same gene in human muscle and to the known role of the IAC in muscle adaptation to mechanical load. In this regard the authors finding of a more pronounced effect of pix-1 mutation or overexpression on crawling (which is under increased mechanical strain/stress) than with swimming is fascinating and I look forward to future experiments using the pix-1 mutants with the recently published burrowing assay and strength assay both of which revealed defects in worm DMD mutants (I also look forward to future human experiments looking for alterations in ARHGEF7 with use/disuse). Lastly, I look forward to future experiments using the pix-1 mutants vs. the worm DMD mutants (e.g. does pix-1 also impact mitochondrial structure/function and/or calcium handling like the DMD mutants (alternatively does pix-1 impact different muscle to muscle channels (for example H channels like in human cardiac muscle) and/or impact pharyngeal muscle (ARHGEF7 has higher expression in human cardiac muscle than in skeletal).

In sum, this is a paper of broad importance across a number of related yet distinct fields that is comprised of well carried out, well reasoned, and nicely complementary experiments that are clearly of interest and immediate translatability into humans.

Minor point:

The authors provide a lot of data that it is the dynamic nature of the PIX-1 GEF function that contributes to the muscle cell boundary IAC defect in pix-1 mutants and have been careful to state “is required for assembly or stability of IACs” in the introduction. Thus, could/should stability appear in the title? (I’m not sure as from a biochemical perspective re-assembly is still assembly)”

Our response: We thank the reviewer for his/her very positive endorsement of our manuscript.

In regards to the minor point, we agree and have changed the title to: “The Rho-GEF PIX-1 Directs Assembly or Stability of Lateral Attachment Structures Between Muscle Cells”

“Reviewer #2 (Remarks to the Author):

C. elegans provides a powerful tool to identify and investigate gene products important for cell adhesion and muscle attachment. This work reports the identification of roles for the exchange factor PIX in integrin-mediated adhesion complexes at muscle borders. PIX had already been implicated in adhesion signaling and in regulation of adhesion turnover, although not specifically in muscle so the current work extends observations to now include C. elegans muscle. Signaling pathways in which PIX operates have also been reported and similar pathways seem to be at play in the muscle. Thus, while this carefully performed study convincingly shows that PIX-1 is important for nematode muscles the advances in mechanistic understanding are very limited. That PIX-1 impacts a subset of IACs is interesting and surprising but again the basis for this is not explored and the general interest of the message of this report seems small.”

Our response: Respectfully, we disagree. We might be wrong, but as far as we know, our results are the first to show that a PIX protein and the PIX pathway have a role in muscle in any animal. The reviewer seems to acknowledge this by stating, “...although not specifically in muscle.” But then, in the next sentence, seems to contradict him/herself: “Signaling pathways in which PIX operates have also been reported and similar pathways seem to be at play in the muscle.” We are not aware of such studies.

“Screening of mutant strains reveals a role for PIX-1 in assembly of muscle attachment sites at the muscle boundary and the absence of functional PIX-1 perturbs assembly of these structures, apparently without altering other adhesion structures in the muscle. The ability of PIX-1 re-expression on the mutant background to restore IAC localization strongly supports a role for PIX in this process. The authors infer that this loss of adhesion contributes to defective nematode motility possibly through alterations in force transmission (Fig 3) but here the rescue construct fails to rescue motility (despite apparently rescuing adhesion in Fig 1) and the expression of the rescue construct on the wild-type background also impaired motility – presumably without

altering adhesions but this was not assessed. Based on these data it seems difficult to conclude that the PIX effect on adhesions contributes to the motility defect. In this regard it might be interesting to determine whether the PIX mutants that do not alter IACs (Fig 1) exhibit any motility defects.”

Our response: We agree with the reviewer that our previous results were a bit puzzling especially that the transgenic array expressing wild type copies of PIX-1 rescued the muscle attachment site defect, but did not rescue the motility defect, and that when the transgene was expressed in a wild type background the motility defect was also observed. Based on the reviewer’s comment, we examined the muscle boundary in the strain in which the transgene was expressed in a wild type background, and as shown in new Fig. 6c, the boundaries are defective. We also determined that the most likely reason for this reduced motility and the defective boundaries is from overexpression of PIX-1, which we demonstrate in a new Fig. 6a and b, and now write (page 10): “Indeed, quantitative western blotting using an antibody to PIX-1, described in Fig. 8, we found that the integrated array expresses 6 times the amount of PIX-1 as found in wild type (Fig. 6a and b).” We also write in the Discussion (bottom of page 16, top of page 17): “We hypothesize that this reduced motility results from decreased transmission of lateral forces between muscle cells. Interestingly, in addition to deficiency of PIX-1, muscle-specific overexpression of PIX-1 protein also results in decreased locomotion and disrupted muscle cell boundaries. Perhaps these results reflect the requirement for PIX-1 signaling to be set at just the optimal level for proper assembly or maintenance of IACs at the muscle cell boundary.”

We provide additional evidence that the loss of function of *pix-1* contributes to both the boundary defect and the motility defect by assessing the phenotype of a CRISPR/Cas9-repaired *pix-1* nonsense mutant (page 10): “Finally, although we found a motility defect in 3 independently generated loss of function mutants, we were concerned that the motility defect might result from mutation in a gene closely linked to *pix-1*. In order to eliminate this possibility, we used CRISPR/Cas9 to correct the TAA stop codon in *pix-1(gk299374)*, to the wild type sequence of a CAA Gln codon. As shown in Fig. 5 c and d, this strain, *pix-1(syb2137gk299374)* displays normal swimming and crawling motility, and as shown in Fig. 7, normal muscle cell boundaries.”

We performed swimming and crawling assays on *gk893650* which carries the P190S mutation and as now show in Fig. 5 and b, the motility of this mutant is not different from wild type. Although this result is somewhat surprising, we now point out that this mutant has a more subtle boundary defect in which the two sides of the “zipper” are separated into 2 lines, and PIX-1 protein can be detected by immunostaining (and is compatible with the normal level of P190S protein by Western blot).

“That PIX-1 localizes to all three IACs in muscle cells seems convincing but the conclusion that the P190S mutant does so is much less convincing from the image shown in Fig 4. Is it possible that it localizes to M lines and dense bodies but not boundaries? Does the 893650 mutant impair localization of integrin, kindlin etc. boundaries (like the deletion mutants do)?”

Our response: In response to these excellent points, we have done additional experiments with the P190S mutant. First, we have obtained better imaging of PAT-6 staining of P190S, and as shown at the bottom of the new Fig. 8c, we observe PAT-6 at M-lines, dense bodies, and at the muscle cell boundary in an characteristic pattern—instead of one line of staining as in wild type, there are two lines, as if the “zipper” has been opened. We can now also observe this double line staining of the IAC component UNC-112 (kindlin) which is shown at the right side of new Fig. 11c. We have also substituted a somewhat better image of PIX-1 staining of P190S (bottom right of new Fig. 8c), which does show localization at all three sites, but more weakly than for wild type. Nevertheless, despite our best efforts, this antibody stains fairly weakly even with wild type muscle. In response to the reviewer’s question, we also present localization of UNC-52(perlecan), PAT-3(β -integrin) and UNC-112 (kindlin) in new Fig. 11c. We observe that the localization of UNC-52 and PAT-3 are impaired like the deletion mutants, but UNC-112 (like PAT-6 and UNC-95) are less disrupted in P190S.

“Some of the data on PIX-1 protein levels are difficult to rationalize – while the effect of the scaffolding protein GIT-1 is reasonable it is surprising that PAK1 but not ced-10 is also required. It would be interesting to probe these results further to attempt to understand why the PAK but not its upstream activator is important.”

Our response: The data is what we found. We think that the data can be rationalized, and is compatible with what has been reported for the PIX pathway in mammals, as we did in the original manuscript: “This data suggests that GIT-1 and PAK-1 are required for the stabilization of PIX-1. The GIT-1 result is consistent with the known high-affinity complex formed between PIX and GIT proteins in mammals (Schlenker et al., 2009). It is also consistent with the reports that GIT1 deficient mice have reduced levels of α -PIX and β -PIX in the brain (Won et al., 2011), and that GIT2 deficient mice have reduced levels of α -PIX in immune cells (Hao et al., 2015). Our finding that PIX-1 is reduced in a *pak-1* mutant is consistent with our finding that by yeast 2 hybrid assays, PIX-1 interacts with PAK-1 (data not shown). It is also compatible with the known interaction of the SH3 domain of β -PIX with PAK1-3 in mammals (Manser et al., 1998).” In our opinion the fact that the loss of a PIX-1 substrate, CED-10 (Rac), does not affect the RacGEF PIX-1, seems reasonable.

“The MD simulations of the P190S GIT-1 mutant may offer the starting point for further studies but without a more detailed investigation of the binding interactions or of the GEF activity the modeling predictions have limited utility.”

Our response: We largely agree with this weakness, however, please consider our new data. We note in the manuscript that *C. elegans* has three Rac proteins encoded by separate genes—CED-10, MIG-2 and RAC-2. And, as we showed in the previous manuscript and now in Fig. 9, only CED-10 is required for assembly of IACs at the muscle cell boundary. Therefore, we asked if the activation of CED-10 is reduced in a *pix-1* lof mutant. We made a transgenic line in which HA-CED-10 was expressed in body wall muscle, and adapted the use of a commercial kit to measure Rac1 activation based on the ability of GST-PAK-PBD to pull down activated or GTP bound CED-10. As now shown in a new Fig. 4, less activated CED-10 was pulled out from *pix-*

1(*gk299374*) than from wild type. With an N=3, the mean level of activated CED-10 from *pix-1(gk299374)* was 54% of the level from wild type. This *pix-1* mutant shows no detectable PIX-1 protein by Western and is likely a null. That the amount of activated CED-10 is not zero is likely because of the expression of other Rac GEFs in nematode muscle. In fact, in the Discussion (page 17) we refer to a new Supplementary Table 1 that lists all the RhoGEF domain proteins expressed in *C. elegans* muscle—17 including PIX-1, and at least two of them are known to be RacGEFs.

As we pointed out in the previous manuscript, the conservation of P190 in the RhoGEF domains of PIX proteins suggests that it is functionally important, and that the P190S mutation might reduce its RacGEF activity. Although, for several technical reasons, we have not established in vitro RacGEF activity assays using PIX-1 and CED-10 in our lab, we tested whether there would be decreased CED-10 activation in the muscle of the P190S mutant. To do this, the P190S mutant was crossed into the transgenic line that expresses HA-CED in body wall muscle, and we conducted a Rac activation pulldown assay as we did for the *pix-1* null mutant. As shown in a new Fig. 13, less activated CED-10 was pulled out from P190S than from wild type. With N=3, the mean level of activated CED-10 from P190S was 52.9% of the level from wild type. Therefore, the P190S mutation results in a normal level of PIX-1 protein that has compromised RacGEF activity. In the Discussion on page 20, we state: “By expressing CED-10 (Rac) specifically in muscle, we were able to show that using a pull-down assay, worms carrying this PIX-1 P190S mutation have a ~50% reduction in the level of activated (GTP bound) CED-10(Rac) in muscle. This was approximately the same level of reduction in activated CED-10 that we observed in a *pix-1* null mutant. This suggests that, indeed, the P190S mutation completely inactivates the RacGEF activity of PIX-1. In the future, biochemical GEF assays using purified proteins will be necessary to determine if this prediction is true. Nevertheless, the muscle cell boundary defect of *pix-1* null and P190S mutants are different; whereas the null mutants result in the absence of IACs, the P190S mutant results in a defect in which components of the IAC localize to muscle cell boundaries, but boundary appears “split”, such that the IACs are formed but are less functional, perhaps less able to anchor cells to the ECM. The fact that the P190S mutant shows some IAC formation and an abundant full-length protein, and is still likely to have zero RacGEF activity, suggests that perhaps PIX-1 has a function in addition to its RacGEF activity.”

We hope that the reviewers agree that we have greatly improved our manuscript and now provide sufficient new and interesting information on the PIX family of proteins to warrant publication in *Nature Communications*.

Thank you very much for your consideration.

Sincerely,

Guy M. Benian

Reviewers' Comments:

Reviewer #2:

Remarks to the Author:

This revised manuscript addresses or rebuts many of my prior concerns. The study is well-performed and carefully interpreted. My only substantial concerns remain my prior issues with the general interest of the report and the magnitude of the advances – while I agree that this is the first evidence for PIX in muscle it appears that what it does in muscle is mostly similar to what it is already known to do elsewhere and there seems to be no mechanistic explanation for its selective effect on a subset of adhesions. However, considering the strong enthusiasm of referee 1 this would seem to be best resolved by the editor.

Minor points:

Line 194 & 195: The authors report “After repeating this experiment three times, the mean level of activated CED-10 from pix-1(gk299374) was 54.0% of the level from wild type” Please include an error or measure of variance for these data. Similar measures are needed for the calculations on lines 355 and 356

August 22, 2020

Dear Reviewer,

Thank you for your positive reception to our revised manuscript. Our response to your specific comments follows.

“REVIEWERS' COMMENTS:

Reviewer #2 (Remarks to the Author):

This revised manuscript addresses or rebuts many of my prior concerns. The study is well-performed and carefully interpreted. My only substantial concerns remain my prior issues with the general interest of the report and the magnitude of the advances – while I agree that this is the first evidence for PIX in muscle it appears that what it does in muscle is mostly similar to what it is already known to do elsewhere and there seems to be no mechanistic explanation for its selective effect on a subset of adhesions. However, considering the strong enthusiasm of referee 1 this would seem to be best resolved by the editor.”

We agree with the reviewer that what we have found for PIX in muscle is very similar to what is already known for the function of PIX in other cell types. However, we would like to point out several strengths of our study: (1) As the reviewer recognizes, this will be the first report of the function of a PIX protein in muscle. (2) Our study, in contrast to many other studies on PIX proteins, is based on analysis of a large number of loss of function mutants. This permitted us to discover the functional importance of the highly conserved P190 residue in the RacGEF domain. This also permitted us to find genetic evidence that a PAK is required for the stability of a PIX, supporting the functional importance of the known biochemical interaction of β -PIX with PAK1-3 in mammals. (3) Our finding that similar phenotypes are obtained from deficiency or overexpression of *pix-1* indicates that the level of a PIX protein needs to be tightly controlled in order to promote proper assembly of integrin adhesions. To the best of our knowledge this is a new contribution to our understanding of PIX proteins, or at least the first report using an in vivo system to show this phenomenon. (4) Although we show that overall the known PIX pathway is also used in muscle (Figure 7a), we found some interesting specificities, which may reflect muscle-specific requirements: Although *C. elegans* has 3 Rac proteins, only CED-10, and not MIG-2 or RAC-2 are required for the IACs at the muscle boundaries; similarly, although *C. elegans* has 3 PAK proteins, PAK-1 and PAK-2 are required at muscle boundaries, but MAX-2 is not.

It is true that our data does not explain why there is a selective effect on a subset of adhesions. However, in the revised manuscript that you reviewed, we did address this question, and we discuss two possibilities: (1) One possibility is that the complexes are assembling and dis-assembling at faster rates at the muscle cell boundaries than at the M-lines and dense bodies and thus would be more sensitive to the Rac cycling from inactive to active and back via the PIX pathway. However, we state that, “...our preliminary FRAP experiments with GFP tagged PAT-6

and UNC-97, show no differences in turnover rates at muscle cell boundaries vs. dense bodies, so this idea does not seem viable.” (2) The other possibility, that we favor, is genetic redundancy in the form of additional proteins, that like PIX, are RacGEFs, and are found at M-lines and dense bodies, but not at muscle cell boundaries. We also provide a supplementary table showing that the *C. elegans* proteome has 17 proteins, including PIX-1, that have RhoGEF (DH) domains, and there are at least 3 leading candidates, UIG-1, TIAM-1 and UNC-73. Please let us investigate this interesting question in the future.

“Minor points:

*Line 194 & 195: The authors report “After repeating this experiment three times, the mean level of activated CED-10 from *pix-1(gk299374)* was 54.0% of the level from wild type” Please include an error or measure of variance for these data. Similar measures are needed for the calculations on lines 355 and 356”*

Thank you for pointing out this omission. We now state on page 9: “After repeating this experiment three times, the mean level of activated CED-10 from *pix-1(gk299374)* was 54.0 +/- 7.6% (mean and standard deviation) of the level from wild type.” And similarly, on page 16: “Repeating this experiment three times, the mean level of activated CED-10 from P190S was 52.9 +/- 5.7% (mean and standard deviation) of the level from wild type.”

Sincerely,

Guy Benian